# MIND THE GAP: OFFLINE POLICY OPTIMIZATION FOR IMPERFECT REWARDS

**Jianxiong Li**♠∗**, Xiao Hu**♠∗**, Haoran Xu**♠**, Jingjing Liu**♠**, Xianyuan Zhan**♠,◇†**,**
**Qing-Shan Jia**♠†**& Ya-Qin Zhang**♠†

♠ Tsinghua University, Beijing, China    ◇ Shanghai Artificial Intelligence Laboratory, Shanghai, China
{li-jx21,hu-x21}@mails.tsinghua.edu.cn, zhanxianyuan@air.tsinghua.edu.cn

## ABSTRACT

Reward function is essential in reinforcement learning (RL), serving as the guiding signal to incentivize agents to solve given tasks, however, is also notoriously difficult to design. In many cases, only imperfect rewards are available, which inflicts substantial performance loss for RL agents. In this study, we propose a unified offline policy optimization approach, *RGM (Reward Gap Minimization)*, which can smartly handle diverse types of imperfect rewards. RGM is formulated as a bi-level optimization problem: the upper layer optimizes a reward correction term that performs visitation distribution matching w.r.t. some expert data; the lower layer solves a pessimistic RL problem with the corrected rewards. By exploiting the duality of the lower layer, we derive a tractable algorithm that enables sampled-based learning without any online interactions. Comprehensive experiments demonstrate that RGM achieves superior performance to existing methods under diverse settings of imperfect rewards. Further, RGM can effectively correct wrong or inconsistent rewards against expert preference and retrieve useful information from biased rewards. Code is available at https://github.com/Facebear-ljx/RGM.

## 1 INTRODUCTION

Reward plays an imperative role in every reinforcement learning (RL) problem. It encodes the desired task behaviors, serving as a guiding signal to incentivize agents to learn and solve a given task. As widely recognized in RL studies, a desirable reward function should not only define the task the agent learns to solve, but also offers the "bread crumbs" that allow the agent to efficiently learn to solve the task (Abel et al., 2021; Singh et al., 2009; Sorg, 2011).

However, due to task complexity and human cognitive biases (Hadfield-Menell et al., 2017), accurately describing a complex task using numerical rewards is often difficult or impossible (Abel et al., 2021; Li et al., 2019). In most practical settings, the rewards are typically "imperfect" and hard to be fixed through reward tuning when online interactions are costly or dangerous (Zhan et al., 2022). Such imperfect rewards are widespread in real-world applications and can appear in forms such as *partially correct rewards, sparse rewards, mismatched rewards from other tasks, and completely incorrect rewards* (see Figure 1 for an intuitive illustration ). These rewards either fail to incentivize agents to learn correct behaviors or cannot provide effective signals to speed up the learning process. Consequently, it is of great importance and practical value to devise a versatile method that can perform robust offline policy optimization under diverse settings of imperfect rewards.

Reward shaping (Ng et al., 1999) is the most common approach to tackling imperfect rewards, but it requires tremendous human efforts and numerous online evaluations. Another possible avenue is imitation learning (IL) (Pomerleau, 1988; Kostrikov et al., 2019) or offline inverse reinforcement learning methods (IRL) (Jarboui & Perchet, 2021), by directly imitating or deriving new rewards from expert behaviors. However, these methods heavily depend on the quantity and quality of expert demonstrations and offline datasets, which are often beyond reach in practice. Another key challenge is how to precisely measure the discrepancy between the given reward in the data and the true reward

---

∗Equal contribution.
†Correspondence to Xianyuan Zhan, Qing-Shan Jia and Ya-Qin Zhang

Figure 1: Diverse settings of imperfect rewards.

of the task. As evaluating the learned policy's behavior under a specific reward function through environment interactions becomes impossible under the offline setting, let alone revising the reward.

In this paper, we investigate the challenge of learning effective offline RL policies under imperfect rewards, when environment interactions are not possible. We first formally define the relative gap between the given and perfect rewards based on state-action visitation distribution matching (referred to as *reward gap*), and formulate the problem as a bi-level optimization problem. In the upper layer, the imperfect rewards are adjusted by a reward correction term, which is learned by minimizing the reward gap toward expert behaviors. In the lower layer, we solve a pessimistic RL problem to obtain the optimized policy under the corrected rewards. By exploiting Lagrangian duality of the lower-level problem, the overall optimization procedure can be tractably solved in a fully-offline manner without any online interactions. We call this approach *Reward Gap Minimization* (RGM). Compared to existing methods, RGM can: 1) evaluate and minimize the reward gap without any online interactions; 2) eliminate the strong dependency on human efforts and numerous expert demonstrations; and 3) handle diverse types of reward settings (e.g., perfect, partially correct, sparse, multi-task data sharing, incorrect) in a unified framework for reliable offline policy optimization.

Through extensive experiments on D4RL datasets (Fu et al., 2020), sparse reward tasks, multi-task data sharing tasks and a discrete-space navigation task, we demonstrate that RGM can achieve superior performance across diverse settings of imperfect rewards. Furthermore, we show that RGM effectively corrects wrong/inconsistent rewards against expert preference and effectively retrieves useful information from biased rewards, making it an ideal tool for practical applications where reward functions are difficult to design.

## 2 RELATED WORK ON DIFFERENT REWARD SETTINGS

We here briefly summarize relevant methodological approaches that handle different types of rewards.

**Perfect rewards**. Directly applying offline RL algorithms is a natural choice when rewards are assumed to be perfect for the given task (Fujimoto et al., 2019; Kumar et al., 2019; 2020; Xu et al., 2021; Fujimoto & Gu, 2021; Kostrikov et al., 2021a;b; Xu et al., 2022a; Li et al., 2023; Lee et al., 2021; Bai et al., 2021; Xu et al., 2023). However, specifying perfect rewards requires deep understanding of the task and domain expertise. Even given the perfect rewards, some offline RL methods still need to shift the rewards to achieve the best performance (Kostrikov et al., 2021a; Kumar et al., 2020), which is shown to be equivalent to engineering the initialization of Q-function estimation that encourages conservative exploitation under offline learning (Sun et al., 2022).

**Partially correct rewards**. Reward shaping is the most common approach to handle partially correct rewards, by modifying the original reward function to incorporate task-specific domain knowledge (Dorigo & Colombetti, 1994; Randløv & Alstrøm, 1998; Ng et al., 1999; Marom & Rosman, 2018; Wu & Lin, 2018). However, these approaches follow a trial-and-error paradigm and require tremendous human efforts. Recent approaches such as population-based method (Jaderberg et al., 2019), optimal reward framework (Chentanez et al., 2004; Sorg et al., 2010; Zheng et al., 2018) and automatic reward shaping (Hu et al., 2020; Devidze et al., 2021; Marthi, 2007) can automatically shape the rewards when online interaction is allowed. However, to the best of the authors' knowledge, no reward shaping or correction mechanism exists for offline policy optimization. Researchers have to discard the given imperfect rewards and resort to other stopgaps like offline IL under offline settings.

**Sparse rewards**. Sparse rewards can be seen as a special case of partially correct rewards. The key challenge of offline policy optimization for sparse rewards is how to effectively back-propagate the sparse signals to stitch up suboptimal trajectories (Levine et al., 2020). Recent works (Kostrikov et al., 2021b; Kumar et al., 2020) use reward shaping to densify the sparse rewards for better performance.

However, reward shaping requires online evaluation and tuning, which is not applicable in the offline setting. Currently, few mechanisms are specifically designed for offline RL to handle sparse rewards.

**Imperfect rewards in multi-task data sharing**. Sharing data across different tasks can potentially enhance offline RL performance on a target task by utilizing additional data from other relevant tasks. As the goals of other relevant tasks are different from that of the target task, the rewards designed for other tasks are naturally imperfect for solving the target task. Since directly sharing datasets from other tasks exacerbates the distribution shift in offline RL (Yu et al., 2021; Bai et al., 2023), prior work such as CDS (Yu et al., 2021) shares data relevant to the target task based on learned Q-values, but it requires access to the functional form of the reward for relabling. CDS+UDS (Yu et al., 2022) directly set the shared rewards to zero without reward relabeling to reduce the bias in the shared rewards, but it cannot completely remedy the reward bias.

**Completely incorrect rewards**. When rewards are believed to be totally wrong or missing, researchers typically adopt offline imitation learning (IL) methods. These methods directly mimic the expert from demonstrations without the presence of a reward signal. Among these approaches, behavior cloning (BC) (Pomerleau, 1988; Florence et al., 2022) is the simplest one, but is vulnerable to covariate shift and compounding errors (Rajaraman et al., 2020). Recent works tackle this problem via distribution matching (Jarboui & Perchet, 2021; Kostrikov et al., 2019; Kim et al., 2021; Ma et al., 2022) or using a discriminator to measure the optimal level of the data and further guide policy learning (Zolna et al., 2020; Xu et al., 2022b; Zhang et al., 2022). These approaches all have strong requirements on the size and coverage of the expert datasets, and only try to imitate the expert rather than improve beyond the policies in data via RL based on the underlying reward of the task.

## 3 PRELIMINARIES

**Markov decision process under imperfect rewards**. We consider the typical Markov Decision Process (MDP) setting (Puterman, 2014), which is defined by a tuple $\mathcal{M} := (S, A, r, T, \mu_0, \gamma)$. $S$ and $A$ represent the state and action space, $r : S \times A \to \mathbb{R}$ is the perfect reward function, $T : S \times A \to \Delta(S)$ is the transition dynamics which represents the probability $T(s_{t+1}|s_t, a_t)$ of the transition from state $s_t$ to state $s_{t+1}$ by executing action $a_t$ at timestep $t$. $\mu_0 \in \Delta(S)$ is the distribution of the initial state $s_0$, and $\gamma \in (0, 1)$ is the discount factor.

The perfect reward function $r(s, a)$ encodes the desired behaviors of the task. But in most cases, we only have access to an imperfect human-designed reward function $\tilde{r}(s, a)$, which may not align well with the target task. This leads to a biased MDP $\widetilde{\mathcal{M}} := (S, A, \tilde{r}, T, \mu_0, \gamma)$ as compared to the original MDP $\mathcal{M}$. To remedy the adverse effects of imperfect reward signals, existing offline policy learning studies (Zolna et al., 2020; Xu et al., 2022b; Ma et al., 2022; Kim et al., 2021; Jarboui & Perchet, 2021) introduce additional expert demonstrations $\mathcal{D}^E = \left\{ \left( s_0^E, a_0^E, s_1^E, \cdots \right)^{(i)} \right\}_{i=0}^{N^E}$ to provide extra information on the desired policy behaviors. We follow a similar setup, but only consume very limited expert demonstrations. In our offline policy optimization setting, we are given a pre-collected dataset $\mathcal{D} = \left\{ (s_0, a_0, \tilde{r}_0, s_1, \cdots)^{(i)} \right\}_{i=0}^N$ that is generated by an unknown behavior policy $\pi^\beta$ and annotated with imperfect rewards $\tilde{r}$. We aim to learn an effective policy $\pi_r : S \to \Delta(A)$ to capture the optimized agent behavior in $\mathcal{M}$ rather than $\widetilde{\mathcal{M}}$ using both $\mathcal{D}$ and a very small expert dataset $\mathcal{D}^E$.

**Reinforcement learning**. Given a MDP and the reward function $r(s, a)$, the goal of RL is to find an optimized policy $\pi_r^*$ to maximize the expected cumulative discount reward: $\pi_r^* = \arg\max_{\pi_r}(1 - \gamma)\mathbb{E}[\sum_{t=0}^\infty \gamma^t r(s_t, a_t) | s_0 \sim \mu_0(\cdot), a_t \sim \pi_r(\cdot|s_t), s_{t+1} \sim T(\cdot|s_t, a_t)]$. This optimization objective can be equivalently written into the following succinct form (Puterman, 2014; Nachum et al., 2019b) by defining the normalized discounted state-action visitation distribution $d^{\pi_r}(s, a)$ (in the rest of the paper, we omit "normalized discounted state-action" for brevity unless otherwise specified):

$$\pi_r^* = \arg\max_{\pi_r} \mathbb{E}_{(s,a) \sim d^{\pi_r}} [r(s, a)] \tag{1}$$

$$d^{\pi_r}(s, a) = (1 - \gamma) \sum_{t=0}^\infty \gamma^t \text{Pr}[s_t = s, a_t = a | s_0 \sim \mu_0(\cdot), a_t \sim \pi_r(\cdot|s_t), s_{t+1} \sim T(\cdot|s_t, a_t)]$$

This RL objective is not directly applicable to offline setting, as it is no longer possible to sample from $d^{\pi_r}$ via online interactions, and serious distributional shift (Kumar et al., 2019) may occur without proper data-related regularization when learning from offline datasets. To tackle these problems, several recent works (Nachum et al., 2019b; Nachum & Dai, 2020; Lee et al., 2021) incorporate a regularizer into Eq. (1) to formulate a pessimistic RL framework that is solvable in the offline setting:

$$\pi_r^* = \arg\max_{\pi_r} \mathbb{E}_{(s,a)\sim d^{\pi_r}}[r(s,a)] - \alpha D\left(d^{\pi_r} \| d^{\mathcal{D}}\right) \tag{2}$$

where $d^{\mathcal{D}}$ is the visitation distribution of dataset $\mathcal{D}$, $D(\cdot\|\cdot)$ represents some statistical discrepancy measures and $\alpha > 0$ controls the strength of the regularization.

## 4    REWARD GAP MINIMIZATION

To handle diverse imperfect reward settings, three challenges have to be tackled:

1) *Measure the gap between the given rewards and the underlying unknown perfect rewards*;

2) *Unify different reward settings and bridge the reward gap*;

3) *Perform offline policy optimization using an integrated framework*.

Our solution to these challenges is Reward Gap Minimization (RGM). We formally define the reward gap in the perspective of visitation distribution matching and introduce a correction term to correct the problematic rewards. Then, we model RGM as a bi-level optimization problem, with the upper layer minimizing the reward gap and the lower layer solving a pessimistic RL problem. To derive a tractable algorithm, we leverage Lagrangian duality to eliminate the requirement for online samples.

### 4.1    DEFINITION OF REWARD GAP

As observed in recent literature, some tasks cannot be captured by a numerical Markovian reward function (Abel et al., 2021). Hence, learning an explicit proxy of the perfect reward function and comparing it to the given rewards is unlikely the best option to characterize the reward gap. In this study, we define the reward gap based on the outcome of the learned agent behavior, i.e., from the perspective of visitation distribution matching.

**Definition 1.** *(Reward gap)  Given an arbitrary reward function $\hat{r}(s,a)$ and the visitation distribution $d^*$ of the optimal policy induced from the perfect rewards $r$, the reward gap between $\hat{r}$ and $r$ is:*

$$D_f\left(d^{\pi_{\hat{r}}^*} \| d^*\right) \tag{3}$$

*where $D_f(p\|q) = \mathbb{E}_{z\sim q}\left[f\left(\frac{p(z)}{q(z)}\right)\right]$ is the $f$-divergence between distributions $p$ and $q$, and $d^{\pi_{\hat{r}}^*}$ represents the visitation distribution induced by $\pi_{\hat{r}}^*$, which is derived using Eq. (2) with $\hat{r}$.*

Note that $d^*$ is unobtainable since the perfect reward function is unknown. We can alternatively use the visitation distribution $d^E$ induced by unknown $\pi^E$ in expert demonstrations $\mathcal{D}^E$ to approximate $d^*$. Next, we discuss how to adjust $\hat{r}$ to minimize the reward gap.

### 4.2    BI-LEVEL OPTIMIZATION

**Reward correction.**    In our study, we consider $\hat{r}(s,a) := \tilde{r}(s,a) + \Delta r(s,a,\tilde{r})$, where $\Delta r(s,a,\tilde{r})$ is a learnable reward correction term that is correlated with the given imperfect rewards $\tilde{r}$ in $\mathcal{D}$. The introduction of $\Delta r(s,a,\tilde{r})$ enables us to exploit useful information within the partially correct rewards, while also correcting the wrong or inconsistent reward signals. We can further use it to construct a bi-level optimization formulation for RGM, where the upper-level problem optimizes the reward correction term to minimize the $f$-divergence between $d^{\pi_{\hat{r}}^*}$ and $d^E$, and the lower-level problem solves $\pi_{\hat{r}}^*$ as the optimal policy of a pessimistic RL problem with the corrected rewards:

$$\Delta r^* = \arg\min_{\Delta r} D_f\left(d^{\pi_{\hat{r}}^*} \| d^E\right) \tag{4}$$

$$\text{s.t.} \quad \pi_{\hat{r}}^* = \arg\max_{\pi_{\hat{r}}} \mathbb{E}_{(s,a)\sim d^{\pi_{\hat{r}}}}[\hat{r}(s,a)] - \alpha D_f\left(d^{\pi_{\hat{r}}} \| d^{\mathcal{D}}\right) \tag{5}$$

The above bi-level optimization formulation poses several technical difficulties, stemming from the complexity of deriving $d^{\pi_{\hat{r}}^*}$ from $\pi_{\hat{r}}^*$, as well as the requirement of online samples from $d^{\pi_{\hat{r}}^*}$, which is impossible under the offline setting. In the following, we present reformulations for both lower and upper-level problems, which leads to a tractable form and an easy-to-implement algorithm.

**Reformulation of the lower-level problem**. We first reformulate the lower-level problem by exploiting duality and the Bellman flow constraint (Puterman, 2014).

**Definition 2.** *(Bellman flow constraint) Let $\mathcal{T}_\star d(s) = \sum_{\bar{s}, \bar{a}} T(s|\bar{s}, \bar{a}) d(\bar{s}, \bar{a})$ denote the transpose (or adjoint) transition operator, the Bellman flow constraint for the visitation distribution $d(s, a)$ is:*

$$\sum_a d(s, a) = (1 - \gamma)\mu_0(s) + \gamma \mathcal{T}_\star d(s), \forall s \in \mathcal{S} \tag{6}$$

If $d(s, a) \geq 0$ satisfies the Bellman flow constraint, then $d(s, a)$ is feasible and there is a one-to-one correspondence between $d$ and the related policy $\pi$: i.e., $d$ is the only visitation distribution for policy $\pi(a|s) = \frac{d(s,a)}{\sum_{\bar{a}} d(s,\bar{a})}$, while $\pi$ is the only policy whose visitation distribution is $d$ (for detailed proof see Puterman (2014)). Then, the lower level problem Eq. (5) can be re-written to a constraint maximization problem w.r.t. $d$ in place of $\pi_{\hat{r}}$:

$$d^{\pi_{\hat{r}}^*} = \underset{d \geq 0}{\arg\max} \; \mathbb{E}_{(s,a) \sim d}[\hat{r}(s,a)] - \alpha \mathrm{D}_f\left(d \| d^{\mathcal{D}}\right) \; \text{s.t.} \sum_a d(s,a) = (1 - \gamma)\mu_0(s) + \gamma \mathcal{T}_\star d(s), \forall s \in S \tag{7}$$

The Lagrange dual problem of Eq. (7) is as follow:

$$\min_{V(s)} \max_{d \geq 0} \mathbb{E}_{(s,a) \sim d}[\hat{r}(s,a)] - \alpha \mathrm{D}_f\left(d \| d^{\mathcal{D}}\right) + \sum_s V(s)\left[(1-\gamma)\mu_0(s) + \gamma \mathcal{T}_\star d(s) - \sum_a d(s,a)\right] \tag{8}$$

where $V(s)$ are Lagrange multipliers. Note that the primal problem Eq. (7) is convex w.r.t. $d$, and under a mild assumption (see Assumption 1 in Appendix A.2), the Slater's condition (Boyd et al., 2004) holds, which means by strong duality, we can solve the original primal problem by solving Eq. (8). After rearranging the terms, Eq. (8) can be equivalently written as the following form (see Lemma 2 in Appendix A.2 for detailed deduction):

$$\min_{V(s)} \max_{d \geq 0} (1 - \gamma)\mathbb{E}_{s \sim \mu_0}[V(s)] + \mathbb{E}_{(s,a) \sim d}[\hat{r}(s,a) + \gamma \mathcal{T}V(s,a) - V(s)] - \alpha \mathrm{D}_f\left(d \| d^{\mathcal{D}}\right) \tag{9}$$

in which $\mathcal{T}V(s,a) = \sum_{s'} T(s'|s,a)V(s')$ denotes the transition operator. Next, by exploiting the Fenchel conjugate, we can further transform the minimax problem Eq. (9) into a tractable single-level unconstrained minimization problem (see Proposition 1 in Appendix A.2 for detailed derivation), which eliminates the requirement of online samples:

$$\min_{V(s)} (1 - \gamma)\mathbb{E}_{s \sim \mu_0}[V(s)] + \alpha \, \mathbb{E}_{(s,a) \sim d^{\mathcal{D}}}\left[f_\star(\frac{\hat{r}(s,a) + \gamma \mathcal{T}V(s,a) - V(s)}{\alpha})\right] \tag{10}$$

where $f_\star$ is the Fenchel conjugate of $f$. In the above formulation, the Lagrange multipliers $V(s)$ can be equivalently perceived as some sort of state-value function, which can be learned and optimized via a parameterized neural network, similar to the treatment used in the DICE-family of RL algorithms (Nachum et al., 2019a; Nachum & Dai, 2020).

**Reformulation of the upper-level problem**. Using the property of Fenchel conjugate, the optimal $d^*$ and $V^*$ from the lower level problem satisfy the following nice relationship (see Proposition 2 in Appendix A.3 for details):

$$\frac{d^{\pi_{\hat{r}}^*}(s,a)}{d^{\mathcal{D}}(s,a)} = f_\star'\left(\frac{\hat{r}(s,a) + \gamma \mathcal{T}V^*(s,a) - V^*(s)}{\alpha}\right) \tag{11}$$

Plugging the above equation into Eq. (5), we can obtain a new objective for the upper-level problem:

$$\Delta r^* = \underset{\Delta r}{\arg\min} \; \mathrm{D}_f\left(f_\star'\left(\frac{\hat{r} + \gamma \mathcal{T}V^* - V^*}{\alpha}\right) d^{\mathcal{D}} \| d^E\right) \tag{12}$$

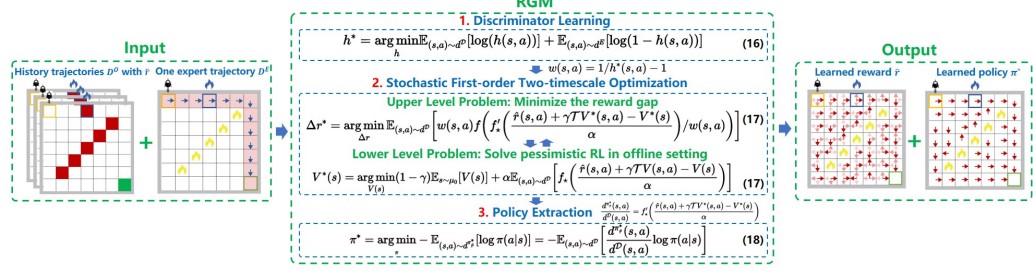

Figure 2: Illustration of the reformulated bi-level optimization problem.

For simplicity, we denote $f'_\star \left( \frac{\hat{r} + \gamma \mathcal{T} V^* - V^*}{\alpha} \right)$ as $g$. By expanding the $f$-divergence, we have:

$$D_f \left( d^\mathcal{D} g \| d^E \right) = \mathbb{E}_{(s,a) \sim d^E} \left[ f \left( \frac{d^\mathcal{D}(s,a) g(s,a)}{d^E(s,a)} \right) \right] = \mathbb{E}_{(s,a) \sim d^\mathcal{D}} \left[ \frac{d^E(s,a)}{d^\mathcal{D}(s,a)} f \left( \frac{d^\mathcal{D}(s,a)}{d^E(s,a)} g(s,a) \right) \right] \quad (13)$$

The above objective involves computing the distribution ratio $w(s,a) \triangleq d^E(s,a)/d^\mathcal{D}(s,a)$. In the tabular case, we can empirically estimate $w(s,a) = \frac{\sum_{(\bar{s},\bar{a}) \in \mathcal{D}^E} \mathbf{1}(\bar{s}=s,\bar{a}=a)/N^E}{\sum_{(\bar{s},\bar{a}) \in \mathcal{D}} \mathbf{1}(\bar{s}=s,\bar{a}=a)/N}$. But in the continuous state-action settings, estimating the distribution ratio $w$ using only samples from $d^\mathcal{D}$ and $d^E$ becomes a challenge. Inspired by previous studies (Goodfellow et al., 2020; Ma et al., 2022), we instead train a discriminator $h : S \times A \rightarrow (0,1)$ to infer if $(s,a)$ samples are from $\mathcal{D}^E$ or not:

$$h^* = \arg \min_h \mathbb{E}_{(s,a) \sim d^\mathcal{D}} \left[ \log(h(s,a)) \right] + \mathbb{E}_{(s,a) \sim d^E} \left[ \log(1 - h(s,a)) \right] \quad (14)$$

where the optimal discriminator is $h^*(s,a) = \frac{d^\mathcal{D}(s,a)}{d^\mathcal{D}(s,a) + d^E(s,a)}$ (Goodfellow et al., 2020). We can optimize the above objective to obtain the optimal $h^*$, and further recover $w(s,a) = 1/h^*(s,a) - 1$.

Finally, combining all the reformulations, the final tractable form of the original bi-level optimization problem Eq. (4)-(5) is given as follows:

$$\Delta r^* = \arg \min_{\Delta r} \mathbb{E}_{(s,a) \sim d^\mathcal{D}} \left[ w(s,a) \cdot f \left( f'_\star \left( \frac{\hat{r}(s,a) + \gamma \mathcal{T} V^*(s,a) - V^*(s)}{\alpha} \right) / w(s,a) \right) \right]$$

$$\text{s.t. } V^*(s) = \arg \min_{V(s)} (1-\gamma) \mathbb{E}_{s \sim \mu_0}[V(s)] + \alpha \mathbb{E}_{(s,a) \sim d^\mathcal{D}} \left[ f_\star \left( \frac{\hat{r}(s,a) + \gamma \mathcal{T} V(s,a) - V(s)}{\alpha} \right) \right] \quad (15)$$

**Policy extraction.** With the learned reward correction term $\Delta r(s, a, \tilde{r})$, we can in principle use existing offline RL algorithms to learn the policy with the corrected rewards. However, this implicates additional policy evaluation and policy improvement steps. A more elegant way is to extract the policy through weighted BC as follows, which is substantially more robust and less expensive:

$$\pi^* = \arg \min_\pi - \mathbb{E}_{(s,a) \sim d^{\pi^*_{\hat{r}}}} [\log \pi(a|s)] = \arg \min_\pi - \mathbb{E}_{(s,a) \sim d^\mathcal{D}} \left[ \frac{d^{\pi^*_{\hat{r}}}(s,a)}{d^\mathcal{D}(s,a)} \log \pi(a|s) \right] \quad (16)$$

where $\frac{d^{\pi^*_{\hat{r}}}(s,a)}{d^\mathcal{D}(s,a)}$ can be calculated from Eq. (11).

### 4.3 PRACTICAL IMPLEMENTATION

In our implementation, we use stochastic first-order two-timescale optimization technique (Borkar, 1997), which has been successfully applied in several RL algorithms (Hong et al., 2020; Cheng et al., 2022), to solve bi-level optimization problems. Specifically, we make the gradient update step size of the upper layer much smaller than the one of the lower layer (see Figure 2 for RGM framework. Refer to Appendix B for additional implementation details of RGM).

## 5 EXPERIMENTS

In this section, we present empirical evaluations of RGM under diverse imperfect reward settings, including partially correct rewards, completely incorrect rewards, sparse rewards, and multi-task data sharing setting on Robomimic (Mandlekar et al., 2021), D4RL-v2 (Fu et al., 2020) and a dataset of a grid-world navigation task. As D4RL MuJoCo tasks are deterministic, we use only one expert trajectory to assist the reward correction and policy learning for these tasks.

Table 1: Average normalized scores of RGM compared with offline IL and RL baselines on D4RL datasets. The scores are from the final 10 evaluations with 5 seeds. (T), (P) and (C) mean policy optimization with true rewards, partially correct rewards and completely incorrect rewards, respectively. "-r","-m","-m-r", and "-m-e" are short for random, medium, medium-replay, and medium-expert, respectively. We obtain the results by running author-provided open-source code, and some scores are reported from TD3+BC and IQL papers. For each dataset, the top 2 scores under partially correct rewards are marked in blue.

| D4RL Dataset | Offline IL | | | Offline RL | | | | | | | | | RGM (T / P / C) | | |
|---|---|---|---|---|---|---|---|---|---|---|---|---|---|---|---|
| | BC | DWBC | SMODICE | TD3+BC (T / P / C) | | | IQL (T / P / C) | | | CQL (T / P / C) | | | | | |
| hopper-r | 4.9 | 23.9 | 5.9 | 8.5 | 13.3 | 0.4 | 7.9 | 1.3 | 0.7 | 8.3 | 1.7 | 0.0 | 29.7 | 21.2 | 25.9 |
| halfcheetah-r | 0.2 | 2.0 | 2.6 | 11.0 | -17.1 | -11.6 | 11.2 | 2.2 | 2.2 | 20.0 | -0.4 | -38.4 | 0.2 | 0.2 | 0.2 |
| walker2d-r | 1.7 | 68.3 | -0.2 | 1.6 | 0.8 | 2.0 | 5.9 | 0.3 | -0.3 | 8.3 | 0.1 | -0.0 | 3.9 | 7.7 | -0.1 |
| hopper-m | 52.9 | 16.5 | 54.5 | 59.3 | 13.7 | 37.3 | 66.2 | 34.0 | 35.5 | 58.5 | 56.4 | 11.2 | 56.2 | 55.5 | 54.6 |
| halfcheetah-m | 42.6 | 8.2 | 42.9 | 48.3 | 35.2 | 1.2 | 47.4 | 42.0 | 35.4 | 44.0 | 43.5 | 4.1 | 40.4 | 40.7 | 40.3 |
| walker2d-m | 75.3 | 18.8 | 1.0 | 83.7 | 30.1 | 17.2 | 78.3 | 68.9 | 22.0 | 72.5 | 71.1 | 3.3 | 73.3 | 72.3 | 38.4 |
| hopper-m-r | 18.1 | 21.4 | 20.4 | 60.9 | 23.5 | 16.3 | 94.7 | 0.7 | 0.7 | 95.0 | 11.5 | 0.0 | 60.3 | 59.1 | 44.5 |
| halfcheetah-m-r | 36.6 | 9.2 | 37.1 | 44.6 | 31.8 | -1.0 | 44.2 | 18.1 | 1.8 | 45.5 | 16.5 | -1.1 | 37.7 | 37.8 | 29.8 |
| walker2d-m-r | 26.0 | 56.6 | 41.1 | 81.8 | 7.8 | -0.7 | 73.8 | 4.9 | -0.2 | 77.2 | 17.4 | -0.0 | 46.3 | 48.6 | 46.1 |
| hopper-m-e | 52.5 | 16.5 | 75.4 | 98.0 | 50.8 | 22.3 | 91.5 | 49.3 | 13.6 | 105.4 | 68.3 | 11.6 | 106.1 | 87.1 | 66.0 |
| halfcheetah-m-e | 55.2 | 0.0 | 88.2 | 90.7 | 35.3 | 1.9 | 86.7 | 53.4 | 35.8 | 91.6 | 64.8 | 11.1 | 85.6 | 81.5 | 78.4 |
| walker2d-m-e | 107.5 | 54.3 | 29.8 | 110.1 | 44.7 | 6.8 | 109.6 | 108.3 | 20.9 | 108.8 | 75.4 | 16.6 | 108.3 | 108.8 | 108.8 |
| Mean Score | 39.5 | 22.6 | 33.2 | 58.2 | 24.6 | 7.7 | 59.8 | 32.0 | 14.0 | 60.5 | 35.5 | -0.3 | 54.1 | 52.0 | 41.9 |

## 5.1 COMPARATIVE RESULTS

**Comparisons for partially correct rewards.** We train RGM and SOTA offline RL methods (TD3+BC (Fujimoto & Gu, 2021), IQL (Kostrikov et al., 2021b) and CQL (Kumar et al., 2020)) under partially correct[1] rewards and report their performances evaluated based on the perfect rewards[2] in Table 1. Table 1 shows that RGM surpasses offline RL methods under partially correct rewards[3] by a large margin and achieves similar performance to offline RL policies that are trained on perfect rewards. This shows a remarkable advantage of RGM as it can alleviate severe performance degradation when perfect rewards are unattainable and hence removes the restrictive requirements on perfect rewards, which can be particularly useful for a wide range of real-world scenarios.

**Comparisons for completely incorrect rewards.** When rewards are believed to be completely incorrect, one generally resorts to IL methods. We compare RGM with BC and SOTA offline IL methods (DWBC (Xu et al., 2022b) and SMODICE (Ma et al., 2022)) that can learn from mixed-quality data. Only offline IL methods are considered as baselines, because other existing methods that tackle incorrect rewards can only be applied in the online settings (see Section 2 for discussions).

In our setting, we train offline IL baselines using the original D4RL dataset $\mathcal{D}$, which may not cover enough expert trajectories. However, DWBC and SMODICE both build on the strong assumption that $\mathcal{D}$ already covers a large proportion of expert datasets, which is a rare case in real scenarios. As a result, Table 1 shows that these two methods suffer from inferior performance when the restrictive requirements on the quality and state-action space coverage of expert data are not satisfied. RGM, however, performs well when nearly no expert trajectories are contained in the offline dataset, because RGM is optimizing an RL objective that relaxes the requirements on the quality of the dataset.

To further illustrate the superiority of RGM, we compare RGM against DWBC and SMODICE under their settings by adding 100~200 expert trajectories into $\mathcal{D}$. Results show that RGM can still outperform SOTA offline IL methods by a large margin (see Table 8 in Appendix D).

**Comparisons for sparse rewards.** We evaluate RGM against BC and offline RL methods TD3+BC, CQL and IQL on Robomimic (Mandlekar et al., 2021) Lift and Can tasks. We also evaluate on the well-known extremely difficult AntMaze tasks. We report the average max success rate as the evaluation metric in Table 2 (See Appendix C.2 for task descriptions and experimental setups).

---

[1] The signs of 50% D4RL rewards are flipped and hence only half rewards can give correct learning signals.

[2] We regard the original D4RL rewards as perfect since we evaluate the policies in terms of these rewards, which can be perceived as solving the tasks encoded in the original D4RL rewards.

[3] All sign of the original rewards is flipped.

Table [2] shows that the offline RL baselines fail miserably on AntMaze tasks[4], as sparse rewards are hard to back-propagate through a very long horizon ($\approx$ 1K steps), while RGM can correctly provide dense signals to guide the ant navigate to the destination. For Robomimic Lift and Can tasks, RGM again outperforms existing methods, while other methods can also achieve reasonable performance. We suspect that these offline datasets may already contain near-optimal trajectories as BC can achieve reasonable performance. Moreover, the planning horizon of both tasks are relatively short ($\approx$ 150 steps), thus is relatively simple for offline RL to back-propagate the sparse signals.

Table 2: Results on sparse reward tasks.

| Dataset | BC | TD3+BC | CQL | IQL | RGM |
|---|---|---|---|---|---|
| Antmaze-m-p | 0 | 0 | 0 | 0 | 13.7 |
| Antmaze-m-d | 0 | 0 | 0 | 0 | 3.3 |
| Lift-MG | 65.3 | 87.3 | 64.0 | 56.0 | 90.3 |
| Can-MG | 64.7 | 55.3 | 64.7 | 50.0 | 66.7 |

**Extension to multi-task data sharing.** We highlight that RGM can also perform well in the offline multi-task data sharing tasks (Yu et al., 2021), which utilize datasets from other relevant tasks to enhance the offline RL performance on a target task. Prior works either require the functional form of rewards to be known for relabeling (Yu et al., 2021) or partially correct the reward biases (Yu et al., 2022). In contrast, RGM systematically corrects the reward biases with-

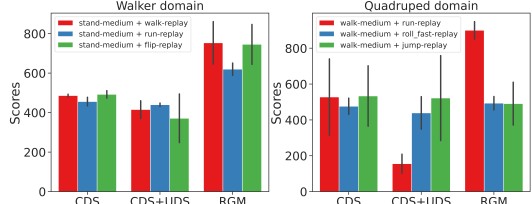

Figure 3: Results on multi-task data sharing tasks.

out reward relabelling, using just one expert trajectory from the target task. To demonstrate the efficacy of RGM compared to SOTA multi-task data sharing algorithms CDS (Yu et al., 2021) and CDS+UDS (Yu et al., 2022), we conduct experiments in multi-task **Walker** (*Stand, Walk, Run, Flip*) and **Quadruped** (*Walk, Run, Roll-Fast, Jump*) domains built on DeepMind Control Suite (Tassa et al., 2018). For each task, we use TD3 (Fujimoto et al., 2018) to collect three types of datasets (*expert, medium, replay*), and share the *replay* dataset of the relevant task with the *medium* dataset of the target task. For RGM, we only draw one expert trajectory for the discriminator training. We report the experimental results in Figure [3], which shows that RGM substantially outperforms CDS and CDS+UDS (see Appendix [C.3] and [D.5] for more experiment details and results).

## 5.2 INVESTIGATIONS ON REWARD CORRECTION

**Benefits of learned rewards.** We investigate the potential benefits of the learned rewards via demonstrative experiments in an 8×8 grid world environment. We observe the learned rewards in RGM enjoy three desirable properties that are unlikely to be provided in other existing methods: 1) *encode long horizon information*; 2) *correct wrong rewards against expert preference*; and 3) *retrieve useful information from existing rewards*, as shown in Figure [4].

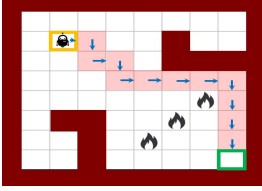
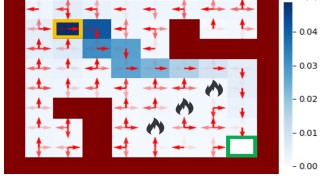
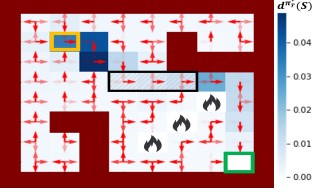

(a) Expert demonstration      (b) Results of zero $\tilde{r}$      (c) Results of partially correct $\tilde{r}$

Figure 4: Learned rewards $\hat{r}$ and optimal distribution $d^{\pi_{\hat{r}}^*}$ trained on two types of imperfect rewards $\tilde{r}$. The opacity of each square represents the value of marginal state distribution $d^{\pi_{\hat{r}}^*}(s)$. The opacity of the arrow shows the learned reward $\hat{r}$, where the darkest arrow points to the direction of the highest reward. The expert starts from ☐, follows the path ▨ and arrow ➡ to reach the goal ☐. $\tilde{r}$ in (b) is +10 at the goal and is zero at other states. $\tilde{r}$ in (c) falsely punishes the agent on ☐ and correctly punishes the RL agent on fire marks 🔥.

---

[4]Note that in IQL and CQL papers, they turn the original sparse rewards into dense rewards by applying the reward subtraction trick (minus 1 on every reward, so the reward becomes negative except at the goal).

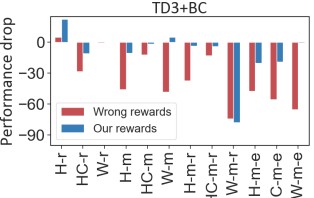 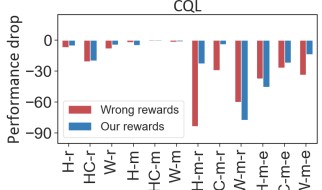 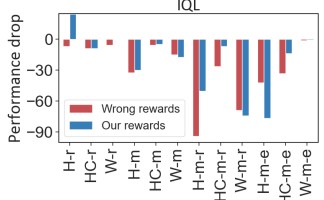

Figure 5: Performance drop of normalized returns of SOTA offline RL methods on D4RL datasets under perfect and RGM corrected rewards. The wrong rewards are the partially correct rewards as in Table 1. *H*: Hopper; *HC*: HalfCheetah; *W*: Walker2d.

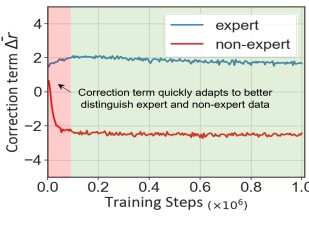

(a) Learning curve of $\Delta r$

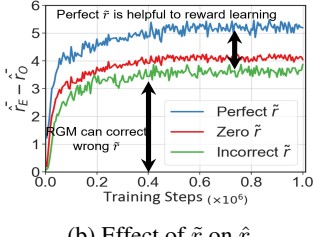

(b) Effect of $\tilde{r}$ on $\hat{r}$

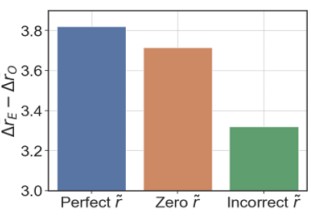

(c) Effect of $\tilde{r}$ on $\Delta r$

Figure 6: Experiments on learned rewards in hopper-m-r task. The superscript "$\bar{}$" denotes the mean value of mini-batch samples. The subscript "E" and "O" denote the value on expert and non-expert data. In (b)(c), large $\bar{\hat{r}}_E - \bar{\hat{r}}_O$ and $\Delta \bar{r}_E - \Delta \bar{r}_O$ indicate that expert and non-expert data are clearly distinguishable according to the learned rewards, and small values mean the opposite.

Specifically, Figure 4b shows that the learned rewards not only recover correct learning signals on the path of the expert, but also generalize well on regions not covered by expert data. In most locations, the agent can navigate to the destination by simply maximizing the one-step reward, meaning that the learned rewards encode long-horizon information. Moreover, Figure 4c shows that the learned rewards can avoid the dangerous fire locations by retrieving useful information provided in imperfect $\tilde{r}$, meanwhile correcting the wrong rewards against expert preference.

**Offline RL with corrected rewards**. The learned corrected rewards $\hat{r}$ obtained by RGM can also be used in other offline RL approaches. To be mentioned, the corrected rewards are optimized based on the specific $\alpha$ in Eq. (5), hence may not be optimal to other offline RL methods. Nevertheless, Figure 5 shows that the corrected rewards can largely remedy the negative effects of the partially correct rewards and even surpass perfect rewards in some datasets.

**Ablations on learned rewards**. Additionally, we investigate the learned rewards in high-dimensional continuous control tasks by inspecting the learning process of both the reward correction term $\Delta r$ and the final learned rewards $\hat{r}$. Figure 6a shows that the reward correction term $\Delta r$ initially cannot distinguish expert and non-expert data well, but adapts and converges quickly. After a few training steps, $\Delta r$ can correctly reward expert data and punish non-expert data very well. We also perform ablations on the effect of diverse types of imperfect rewards $\tilde{r}$ on $\Delta r$ and $\hat{r}$. Figure 6b shows that a perfect $\tilde{r}$ is beneficial to enlarge reward differences on expert and non-expert samples, and incorrect $\tilde{r}$ can be counterproductive. Nevertheless, RGM can largely correct the wrong rewards and produce reasonable learning signals. Similar effects are also observed on $\Delta r$, as Figure 6c shows.

## 6 DISCUSSION AND CONCLUSION

In this paper, we propose RGM (Reward Gap Minimization), a unified offline policy optimization approach applicable to diverse settings of imperfect rewards. RGM is formulated as a bi-level optimization problem, which achieves reward correction and simultaneous policy learning in a fully offline paradigm. Extensive experiments and illustrative examples show that RGM can perform robust policy optimization under imperfect rewards. Several desirable properties are also identified in the corrected rewards learned by RGM. One limitation of RGM is the need for a small expert dataset, which may not be easily accessible in some applications. However, RGM relaxes the strong dependencies on online reward tuning and tedious human efforts, which renders it a powerful tool to solve many real-world problems.

ACKNOWLEDGMENTS

This work is supported by funding from Haomo.AI, and National Natural Science Foundation of China under Grant 62125304, 62073182. The authors would also like to thank the anonymous reviewers for their feedback on the manuscripts.

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

# A    PROOFS

## A.1    BACKGROUND

We begin by briefly introducing the Fenchel conjugate (also known as convex conjugate or Legendre–Fenchel transformation):

**Definition 3.** *(Fenchel conjugate)    In a real Hilbert space $\mathcal{X}$, if a function $f(x)$ is proper, then the Fenchel conjugate $f_\star$ of $f$ at $y$ is:*

$$f_\star(y) = \sup_{x \in \mathcal{X}} (y^T x - f(x)) \tag{17}$$

*where the domain of the $f_\star(y)$ is given by:*

$$\mathrm{dom}\, f_\star = \left\{ y : \sup_{x \in \mathrm{dom}\, f} \left( y^T x - f(x) \right) < \infty \right\} \tag{18}$$

*If $f$ is convex and lower semi-continuous as well, we have the duality $f_{\star\star}(x) = f(x)$. Furthermore, if $f$ is also differentiable, then the maximizer $x^*$ of $f_\star(y)$ satisfies:*

$$x^* = f_\star'(y) \tag{19}$$

Next, we present the interchangeability principle, which plays a key role in Proposition 1.

**Lemma 1.** *(Interchangeability principle)    Let $\xi$ be a random variable on $\Xi$ and assume for any $\xi \in \Xi$, function $g(\cdot, \xi)$ is a proper and upper semi-continuous concave function. Then*

$$\mathbb{E}_\xi \left[ \max_{u \in \mathbb{R}} g(u, \xi) \right] = \max_{u(\cdot) \in \mathcal{G}(\Xi)} \mathbb{E}_\xi [g(u(\xi), \xi)] \tag{20}$$

*where $\mathcal{G}(\Xi) = \{u(\cdot) : \Xi \to \mathbb{R}\}$ is the entire space of functions defined on support $\Xi$.*

*Proof.*  Please refer to (Dai et al., 2017; Rockafellar & Wets, 2009). □

## A.2    PROOF OF TRACTABLE TRANSFORMATION OF THE LOWER-LEVEL PROBLEM

We start our proof from the original bi-level optimization problem Eq. (4) and Eq. (5). Using the Bellman flow constraint for Eq. (5) yields:

$$\Delta r^* = \arg\min_{\Delta r} \, \mathrm{D}_f \left( d^{\pi_{\hat{r}}^*} \| d^E \right)$$
$$\text{s.t. } d^{\pi_{\hat{r}}^*} = \arg\max_{d \geq 0} \, \mathbb{E}_{(s,a) \sim d}[\hat{r}(s,a)] - \alpha \mathrm{D}_f \left( d \| d^{\mathcal{D}} \right) \tag{21}$$
$$\text{s.t. } \sum_a d(s,a) = (1-\gamma)\mu_0(s) + \gamma \mathcal{T}_\star d(s), \forall s \in S$$

**Assumption 1.** *There exists at least one $d$ such that:*

$$\sum_a d(s,a) = (1-\gamma)\mu_0(s) + \gamma \mathcal{T}_\star d(s), \, d(s) > 0, \, \forall s \in S \tag{22}$$

We note that this assumption is mild since when every state is reachable from the initial state distribution, the assumption is satisfied, which is common in practice.

Slater's theorem (Boyd et al., 2004) states that strong duality holds, if the optimization problem is strictly feasible (Slater's condition holds) and the problem is convex. So under Assumption 1 with the fact that the lower level problem is convex w.r.t. $d$, the strong duality holds, which means that the above lower level problem can be re-written as the following form:

$$\min_{V(s)} \max_{d \geq 0} \, \mathbb{E}_{(s,a) \sim d}[\hat{r}(s,a)] - \alpha \mathrm{D}_f \left( d \| d^{\mathcal{D}} \right) + \sum_s V(s) \left[ (1-\gamma)\mu_0(s) + \gamma \mathcal{T}_\star d(s) - \sum_a d(s,a) \right] \tag{23}$$

**Lemma 2.** *The minimax problem:*

$$\min_{V(s)} \max_{d \geq 0} \; \mathbb{E}_{(s,a) \sim d}[\hat{r}(s,a)] - \alpha D_f\left(d\|d^{\mathcal{D}}\right) + \sum_s V(s)\left[(1-\gamma)\mu_0(s) + \gamma \mathcal{T}_\star d(s) - \sum_a d(s,a)\right] \quad (24)$$

*can be equivalently written as:*

$$\min_{V(s)} \max_{d \geq 0} \; (1-\gamma)\mathbb{E}_{s \sim \mu_0}[V(s)] + \mathbb{E}_{(s,a) \sim d}\left[\hat{r}(s,a) + \gamma \mathcal{T}V(s,a) - V(s))\right] - \alpha D_f\left(d\|d^{\mathcal{D}}\right) \quad (25)$$

*Proof.*

$$\mathbb{E}_{(s,a) \sim d}[\hat{r}(s,a)] - \alpha D_f\left(d\|d^{\mathcal{D}}\right) + \sum_s V(s)\left[(1-\gamma)\mu_0(s) + \gamma \mathcal{T}_\star d(s) - \sum_a d(s,a)\right]$$

$$= \mathbb{E}_{(s,a) \sim d}[\hat{r}(s,a)] - \alpha D_f\left(d\|d^{\mathcal{D}}\right) + \sum_s V(s)\left[(1-\gamma)\mu_0(s) + \gamma \sum_{\bar{s},\bar{a}} T(s|\bar{s},\bar{a})d(\bar{s},\bar{a}) - \sum_a d(s,a)\right]$$

$$= \sum_{s,a} d(s,a)\hat{r}(s,a) - \alpha D_f\left(d\|d^{\mathcal{D}}\right) + (1-\gamma)\sum_s \mu_0(s)V(s) + \gamma \sum_{\bar{s},\bar{a}} d(\bar{s},\bar{a})\sum_s T(s|\bar{s},\bar{a})V(s) - \sum_{s,a} d(s,a)V(s)$$

$$= \sum_{s,a} d(s,a)\hat{r}(s,a) - \alpha D_f\left(d\|d^{\mathcal{D}}\right) + (1-\gamma)\sum_s \mu_0(s)V(s) + \gamma \sum_{s,a} d(s,a)\sum_{s'} T(s'|s,a)V(s') - \sum_{s,a} d(s,a)V(s)$$

$$= (1-\gamma)\sum_s \mu_0(s)V(s) + \sum_{s,a} d(s,a)\left[\hat{r}(s,a) + \gamma \sum_{s'} T(s'|s,a)V(s') - V(s)\right] - \alpha D_f\left(d\|d^{\mathcal{D}}\right)$$

$$= (1-\gamma)\mathbb{E}_{s \sim \mu_0}[V(s)] + \mathbb{E}_{(s,a) \sim d}\left[\hat{r}(s,a) + \gamma \mathcal{T}V(s,a) - V(s))\right] - \alpha D_f\left(d\|d^{\mathcal{D}}\right)$$

$$(26)$$

$\square$

**Proposition 1.** *The minimax problem:*

$$\min_{V(s)} \max_{d \geq 0} \; \mathbb{E}_{(s,a) \sim d}[\hat{r}(s,a)] - \alpha D_f\left(d\|d^{\mathcal{D}}\right) + \sum_s V(s)\left[(1-\gamma)\mu_0(s) + \gamma \mathcal{T}_\star d(s) - \sum_a d(s,a)\right] \quad (27)$$

*shares the same optimal value as the following minimization problem:*

$$\min_{V(s)} (1-\gamma)\mathbb{E}_{s \sim \mu_0}[V(s)] + \alpha \, \mathbb{E}_{(s,a) \sim d^{\mathcal{D}}}\left[f_\star\left(\frac{\hat{r}(s,a) + \gamma \mathcal{T}V(s,a) - V(s)}{\alpha}\right)\right] \quad (28)$$

*where $f_\star$ is the Fenchel conjugate function of $f$ with $\mathrm{dom}\, f = \{u : u \geq 0\}$*

*Proof.*    Using Lemma 2, this minimax problem can be re-written as:

$$\min_{V(s)} \max_{d \geq 0} (1-\gamma)\mathbb{E}_{s \sim \mu_0}[V(s)] + \mathbb{E}_{(s,a) \sim d}\left[\hat{r}(s,a) + \gamma \mathcal{T}V(s,a) - V(s)\right] - \alpha D_f\left(d\|d^{\mathcal{D}}\right) \quad (29)$$

Next,

$$\min_{V(s)} \max_{d \geq 0} (1-\gamma)\mathbb{E}_{s \sim \mu_0}[V(s)] + \mathbb{E}_{(s,a) \sim d}\left[\hat{r}(s,a) + \gamma \mathcal{T}V(s,a) - V(s)\right] - \alpha D_f\left(d\|d^{\mathcal{D}}\right)$$

$$= \min_{V(s)} (1-\gamma)\mathbb{E}_{s \sim \mu_0}[V(s)] + \max_{d \geq 0} \mathbb{E}_{(s,a) \sim d}\left[\hat{r}(s,a) + \gamma \mathcal{T}V(s,a) - V(s)\right] - \alpha D_f\left(d\|d^{\mathcal{D}}\right)$$

$$= \min_{V(s)} (1-\gamma)\mathbb{E}_{s \sim \mu_0}[V(s)] + \alpha \underbrace{\left[\max_{d \geq 0} \mathbb{E}_{(s,a) \sim d}\left[\frac{\hat{r}(s,a) + \gamma \mathcal{T}V(s,a) - V(s)}{\alpha}\right] - D_f\left(d\|d^{\mathcal{D}}\right)\right]}_{L}$$

$$(30)$$

$L$ in the last step reduces to:

$$
\alpha \left[ \max_{d \geq 0} \mathbb{E}_{(s,a) \sim d} \left[ \frac{\hat{r}(s,a) + \gamma \mathcal{T} V(s,a) - V(s)}{\alpha} \right] - \mathrm{D}_f \left( d \| d^{\mathcal{D}} \right) \right]
$$

$$
= \alpha \left[ \max_{d \geq 0} \mathbb{E}_{(s,a) \sim d^{\mathcal{D}}} \left[ \frac{d(s,a)}{d^{\mathcal{D}}(s,a)} \frac{(\hat{r}(s,a) + \gamma \mathcal{T} V(s,a) - V(s))}{\alpha} \right] - \mathbb{E}_{(s,a) \sim d^{\mathcal{D}}} \left[ f \left( \frac{d(s,a)}{d^{\mathcal{D}}(s,a)} \right) \right] \right]
$$

$$
= \alpha \left[ \max_{d \geq 0} \mathbb{E}_{(s,a) \sim d^{\mathcal{D}}} \left[ \frac{d(s,a)}{d^{\mathcal{D}}(s,a)} \frac{(\hat{r}(s,a) + \gamma \mathcal{T} V(s,a) - V(s))}{\alpha} - f \left( \frac{d(s,a)}{d^{\mathcal{D}}(s,a)} \right) \right] \right]
$$

$$
= \alpha \, \mathbb{E}_{(s,a) \sim d^{\mathcal{D}}} \left[ \max_{d(s,a) \geq 0} \frac{d(s,a)}{d^{\mathcal{D}}(s,a)} \frac{(\hat{r}(s,a) + \gamma \mathcal{T} V(s,a) - V(s))}{\alpha} - f \left( \frac{d(s,a)}{d^{\mathcal{D}}(s,a)} \right) \right] \tag{31}
$$

$$
= \alpha \, \mathbb{E}_{(s,a) \sim d^{\mathcal{D}}} \left[ \max_{\frac{d(s,a)}{d^{\mathcal{D}}(s,a)} \geq 0} \frac{d(s,a)}{d^{\mathcal{D}}(s,a)} y(s,a) - f \left( \frac{d(s,a)}{d^{\mathcal{D}}(s,a)} \right) \right]
$$

$$
= \alpha \, \mathbb{E}_{(s,a) \sim d^{\mathcal{D}}} \left[ f_\star (y(s,a)) \right]
$$

where $y(s,a) = \frac{\hat{r}(s,a) + \gamma \mathcal{T} V(s,a) - V(s)}{\alpha}$, the third step follows the interchangeability principle (Lemma 1) and the last step comes from the Fenchel conjugate of convex function $f$ [5].  $\square$

Using this result, we finally yield the tractable lower-level problem Eq. (10).

### A.3 PROOF OF TRACTABLE TRANSFORMATION OF THE UPPER-LEVEL PROBLEM

**Proposition 2.** *The original upper-level problem*

$$
\min_{\Delta r} D_f \left( d^{\pi_{\hat{r}}^*} \| d^E \right) \tag{32}
$$

*can be equivalently written as:*

$$
\min_{\Delta r} D_f \left( f'_\star \left( \frac{\hat{r} + \gamma \mathcal{T} V^* - V^*}{\alpha} \right) d^{\mathcal{D}} \| d^E \right) \tag{33}
$$

*where $d^{\pi_{\hat{r}}^*}$ is the optimal state-action visitation distribution of Eq. (7)*

*Proof.* By the property Eq. (19), the maximizer $\left( \frac{d(s,a)}{d^{\mathcal{D}}(s,a)} \right)^*$ of $f_\star(y(s,a))$ in Eq. (31) satisfies

$$
\left( \frac{d(s,a)}{d^{\mathcal{D}}(s,a)} \right)^* = f'_\star \left( \frac{\hat{r}(s,a) + \gamma \mathcal{T} V(s,a) - V(s)}{\alpha} \right) \tag{34}
$$

Given $V^*$, we have:

$$
\frac{d^{\pi_{\hat{r}}^*}(s,a)}{d^{\mathcal{D}}(s,a)} = f'_\star \left( \frac{\hat{r}(s,a) + \gamma \mathcal{T} V^*(s,a) - V^*(s)}{\alpha} \right) \tag{35}
$$

Substituting this result into the original upper-level problem completes the proof.  $\square$

Next, we denote $f'_\star \left( \frac{\hat{r} + \gamma \mathcal{T} V^* - V^*}{\alpha} \right)$ as $g$. By expanding the $f$-divergence, we have the upper-level objective:

$$
\mathrm{D}_f \left( d^{\mathcal{D}} g \| d^E \right) = \mathbb{E}_{(s,a) \sim d^E} \left[ f \left( \frac{d^{\mathcal{D}}(s,a) g(s,a)}{d^E(s,a)} \right) \right] \tag{36}
$$

$$
= \mathbb{E}_{(s,a) \sim d^{\mathcal{D}}} \left[ \frac{d^E(s,a)}{d^{\mathcal{D}}(s,a)} f \left( \frac{d^{\mathcal{D}}(s,a)}{d^E(s,a)} g(s,a) \right) \right] \tag{37}
$$

$$
= \mathbb{E}_{(s,a) \sim d^{\mathcal{D}}} \left[ w(s,a) f \left( \frac{g(s,a)}{w(s,a)} \right) \right] \tag{38}
$$

---

[5]$\mathrm{dom} \, f = \{ u : u \geq 0 \}$ and $f$ is convex, so $f_\star(y) = -f(0)$ when $y \leq f'(0)$.

where the distribution ratio $w(s, a) \triangleq d^E(s, a)/d^{\mathcal{D}}(s, a)$.

Finally, by combining proposition 1 and proposition 2, the original bi-level optimization problem Eq. (4)-(5) is rewritten equivalently as follows:

$$\Delta r^* = \arg\min_{\Delta r} \mathbb{E}_{(s,a) \sim d^{\mathcal{D}}} \left[ w(s, a) f \left( f'_\star \left( \frac{\hat{r}(s, a) + \gamma \mathcal{T} V^*(s, a) - V^*(s)}{\alpha} \right) / w(s, a) \right) \right]$$

$$\text{s.t. } V^*(s) = \arg\min_{V(s)} (1 - \gamma) \mathbb{E}_{s \sim \mu_0}[V(s)] + \alpha \, \mathbb{E}_{(s,a) \sim d^{\mathcal{D}}} \left[ f_\star \left( \frac{\hat{r}(s, a) + \gamma \mathcal{T} V(s, a) - V(s)}{\alpha} \right) \right]$$
$$(39)$$

# B    IMPLEMENTATION DETAILS OF RGM

## B.1    RGM WITH KL-DIVERGENCE

In this section, we introduce the implementation details of RGM. For KL-divergence, we have $f(x) = x \log x$ and its Fenchel conjugate is $f_\star(x) = e^{x-1}$. However, this exponential form is numerically unstable and prone to value explosion in practice. We address this issue by using the fact that the conjugate of the negative entropy function, restricted to the probability simplex, is the log-sum-exp function (Boyd et al., 2004), i.e., $D_{\star,f}(y) = \log \mathbb{E}_{x \sim q}[\exp y(x)]$. Then, the optimization problem of RGM with KL divergence is

$$\min_{\Delta r} \mathbb{E}_{(s,a) \sim d^{\mathcal{D}}} \left[ \text{Softmax}\left( \frac{\text{Adv}(\Delta r, V^*)}{\alpha} \right) \left( \log \frac{d^{\mathcal{D}}(s, a)}{d^E(s, a)} + \log \left( \text{Softmax}\left( \frac{\text{Adv}(\Delta r, V^*)}{\alpha} \right) \right) \right) \right]$$

$$\text{s.t.} V^* = \arg\min_V (1 - \gamma) \mathbb{E}_{s \sim \mu_0}[V(s)] + \alpha \log \mathbb{E}_{(s,a) \sim d^{\mathcal{D}}} \left[ \exp\left( \frac{\text{Adv}(\Delta r, V)}{\alpha} \right) \right]$$
$$(40)$$

where, $\text{Adv}(\Delta r, V) := \hat{r}(s, a) + \gamma \mathcal{T} V(s, a) - V(s) = \tilde{r}(s, a) + \Delta r(s, a, \tilde{r}) + \gamma \mathcal{T} V(s, a) - V(s)$ and $\log \frac{d^{\mathcal{D}}(s,a)}{d^E(s,a)}$ can be obtained by training a discriminator $\log \frac{d^{\mathcal{D}}(s,a)}{d^E(s,a)} = -\log\left(\frac{1}{h^*} - 1\right)$ using Eq. (14) in continuous MDPs. The importance ratio used to extract the policy is

$$\psi^*(s, a) = \frac{d^{\pi^*_{\hat{r}}}(s, a)}{d^{\mathcal{D}}(s, a)} = \text{Softmax}\left[ \frac{\tilde{r} + \Delta r + \gamma \mathcal{T} V^*(s, a) - V^*(s)}{\alpha} \right] \tag{41}$$

### B.1.1    OPTIMIZE WITHOUT SUM-EXP

Note that in the upper level objective of Eq. (40), we need to calculate a log-sum-exp value in the denominator of the log(Softmax) term, where $\log\left(\text{Softmax}(\text{Adv}(\Delta r, V^*)/\alpha)\right) = \text{Adv}(\Delta r, V^*)/\alpha - \log \sum_{s,a \in \mathcal{S} \times \mathcal{A}} \exp(\text{Adv}(\Delta r, V^*)/\alpha)$. In low-dimensional discrete state-action space, we can easily get this value via summing over the overall space. In high-dimensional continuous MDPs, however, it is pretty difficult to retrieve the value because it requires integration over the entire space. CQL (Kumar et al., 2020) approximates this value via importance sampling but requires additional samples from the entire state-action space. There are some other methods like Markov Chain Monte Carlo (MCMC) or Score Match (SM) (Song & Kingma, 2021) that can approximate the update gradient but bring additional computation costs and suffer from some technical issues.

Fortunately, we can subtly circumvent the log-sum-exp term by optimizing the upper bound of the original upper-level problem using the following inequality (Boyd et al., 2004):

$$\max_{x_i \in \mathcal{B}}\{x_1, ..., x_n\} \leq \max\{x_1, ..., x_n\} \leq \log \sum_i^n \exp(x_i) \tag{42}$$

where $\max_{x_i \in \mathcal{B}}\{x_1, ..., x_n\}$ is the max value in a mini-batch $\mathcal{B}$ which is sampled from $\{x_1, ..., x_n\}$. For simplicity, we denote $\max_{x_i \in \mathcal{B}}\{x_1, ..., x_n\}$ as $\max_{x_i \in \mathcal{B}}\{\boldsymbol{x}\}$. Substituting Eq. (42) into the upper-level problem of Eq. (40), we get the upper bound of the original upper-level optimization objective:

$$\text{Upper}(40) = \mathbb{E}_{(s,a)\sim d^{\mathcal{D}}}\left[\text{Softmax}\left(\frac{\text{Adv}(\Delta r, V^*)}{\alpha}\right)\left(\log\frac{d^{\mathcal{D}}(s,a)}{d^E(s,a)} + \frac{\text{Adv}(\Delta r, V^*)}{\alpha} - \log\sum_{s,a\in\mathcal{S}\times\mathcal{A}}\exp\left(\frac{\text{Adv}(\Delta r, V^*)}{\alpha}\right)\right)\right]$$

$$\leq \mathbb{E}_{(s,a)\sim d^{\mathcal{D}}}\left[\text{Softmax}\left(\frac{\text{Adv}(\Delta r, V^*)}{\alpha}\right)\left(\log\frac{d^{\mathcal{D}}(s,a)}{d^E(s,a)} + \frac{\text{Adv}(\Delta r, V^*)}{\alpha} - \max_{\mathcal{B}}\left\{\frac{\boldsymbol{\text{Adv}(\Delta r, V^*)}}{\alpha}\right\}\right)\right]$$

$$\propto \mathbb{E}_{(s,a)\sim d^{\mathcal{D}}}\left[\exp\left(\frac{\text{Adv}(\Delta r, V^*)}{\alpha}\right)\left(\log\frac{d^{\mathcal{D}}(s,a)}{d^E(s,a)} + \frac{\text{Adv}(\Delta r, V^*)}{\alpha} - \max_{\mathcal{B}}\left\{\frac{\boldsymbol{\text{Adv}(\Delta r, V^*)}}{\alpha}\right\}\right)\right]$$

$$(43)$$

where $\text{Upper}(40)$ denotes the upper level objective in Eq. (40).

Replacing Eq. (43) to the upper level objective in Eq. (40), we obtain the final optimization problem:

$$\min_{\Delta r}\mathbb{E}_{(s,a)\sim d^{\mathcal{D}}}\left[\exp\left(\frac{\text{Adv}(\Delta r, V^*)}{\alpha}\right)\left(\log\frac{d^{\mathcal{D}}(s,a)}{d^E(s,a)} + \frac{\text{Adv}(\Delta r, V^*)}{\alpha} - \max_{\mathcal{B}}\left\{\frac{\boldsymbol{\text{Adv}(\Delta r, V^*)}}{\alpha}\right\}\right)\right]$$

$$\text{s.t.} V^* = \arg\min_V (1-\gamma)\mathbb{E}_{s\sim\mu_0}[V(s)] + \alpha\log\mathbb{E}_{(s,a)\sim d^{\mathcal{D}}}\left[\exp\left(\frac{\text{Adv}(\Delta r, V^*)}{\alpha}\right)\right]$$

$$(44)$$

We practically utilize the same mini-batch $\mathcal{B}$ as that of SGD gradient update step to calculate $\max_{\mathcal{B}}\left\{\frac{\boldsymbol{\text{Adv}(\Delta r, V^*)}}{\alpha}\right\}$. Note that the exp term in the upper-level problem is prone to value explosion in practice, we clip the exp value to $(-\infty, 100]$ like IQL (Kostrikov et al., 2021b) does to improve training stability.

When extracting the policy, we can ignore the annoying sum-exp term in the denominator of Softmax and get the following ratio, because it does not influence the direction of gradients to update the policy.

$$\psi^*(s,a) = \frac{d^{\pi^*_{\hat{r}}}(s,a)}{d^{\mathcal{D}}(s,a)} \propto \exp\left[\frac{\tilde{r} + \Delta r + \gamma\mathcal{T}V^*(s,a) - V^*(s)}{\alpha}\right] := \tilde{\psi}^*(s,a) \qquad (45)$$

However, using Eq.(45), we can only get an unnormalized distribution ratio instead of an exact one. We resort to self-normalized importance sampling (Owen, 2013) to obtain a normalized ratio:

$$\psi^*(s,a) = \frac{\tilde{\psi}^*(s,a)}{\mathbb{E}_{(s,a)\sim d^{\mathcal{D}}}[\tilde{\psi}^*(s,a)]} \qquad (46)$$

## B.2 RGM WITH $\mathcal{X}^2$-DIVERGENCE

Additionally, we can also implement RGM using $\mathcal{X}^2$-divergence. For $\mathcal{X}^2$-divergence, we have $f(x) = \frac{1}{2}(x-1)^2$ with $\text{dom} f = \{x : x \geq 0\}$[6] and its Fenchel conjugate is $f_\star(x) = \frac{1}{2}(x+1)^2$ and $f'_\star(x) = \max(0, x+1)$. Then, the optimization objective of RGM with $\mathcal{X}^2$ divergence is

$$\min_{\Delta r}\mathbb{E}_{(s,a)\sim d^{\mathcal{D}}}\left[\frac{d^E(s,a)}{2d^{\mathcal{D}}(s,a)}\left(\max\left(0, \frac{\text{Adv}(\Delta r, V^*)}{\alpha} + 1\right)\frac{d^{\mathcal{D}}(s,a)}{d^E(s,a)} - 1\right)^2\right]$$

$$\text{s.t } V^* = \arg\min_V (1-\gamma)\mathbb{E}_{s\sim\mu_0}[V(s)] + \frac{\alpha}{2}\mathbb{E}_{(s,a)\sim d^{\mathcal{D}}}\left[\left(\frac{\text{Adv}(\Delta r, V)}{\alpha}\right)^2\right]$$

$$(47)$$

The importance ratio used to extract the policy is:

$$\psi^*(s,a) = \frac{d^{\pi^*_{\hat{r}}}(s,a)}{d^{\mathcal{D}}(s,a)} = \max\left(0, \frac{\tilde{r} + \Delta r + \gamma\mathcal{T}V^*(s,a) - V^*(s)}{\alpha} + 1\right) \qquad (48)$$

For RGM with KL-divergence, the upper layer contains an exponential term $\exp(\frac{\text{Adv}(\delta r, V^*)}{\alpha})$, which may pose numerical instability. For RGM with $\chi^2$ divergence, $f'_\star(x) = \max(0, x+1)$ and so the

---

[6]On account of the state-action visitation distribution $d \geq 0$

gradient vanishes when $x + 1 < 0$, which makes the policy learning slow or even fail. In practice, we follow the criteria from SMODICE (Ma et al., 2022) by monitoring the initial policy loss to choose the types of $f$-divergence.

### B.3 RGM HYPERPARAMETERS AND PSEUDOCODE

For continuous MDPs with high dimensional state-action spaces, we implement RGM by parameterizing $h_\tau, \Delta r_\phi, V_\theta$ and $\pi_w$ using deep neural networks with parameter $\tau, \phi, \theta$ and $w$, respectively. We implement RGM based on a two-time scale first-order stochastic gradient update, where the reward correction term is updated much slower than the Lagrangian multiplier $V$. We choose the cosine annealing learning rate schedule of the reward correction term and policy network to stabilize the training process. To make the reward correction term comparable w.r.t the original imperfect rewards, we normalize the imperfect rewards to standard Gaussian distribution $\mathcal{N}(0, 1)$ and strict the output range of $\Delta r_\phi$ to $[-3, 3]$ by Tanh function. The conclusive hyperparameters can be found in Table 3.

Table 3: The hyperparameters of RGM with deep neural networks

|  | Hyperparameter | Value |
|---|---|---|
| Architecture | Reward correction hidden dim | 256 |
|  | Reward correction layers | 2 |
|  | Reward correction activation function | ReLU |
|  | Discriminator hidden dim | 512 |
|  | Discriminator layers | 3 |
|  | Discriminator activation function | Tanh |
|  | $V$ hidden dim | 256 |
|  | $V$ hidden layers | 2 |
|  | $V$ activation function | ReLU |
|  | Policy hidden dim | 256 |
|  | Policy hidden layers | 2 |
|  | Policy activation function | ReLU |
| RGM Hyperparameters | Optimizer | Adam (Kingma & Ba, 2015) |
|  | Reward correction learning rate $l_\phi$ | 3e-7 |
|  | Reward correction learning rate schedule | cosine annealing |
|  | Discriminator learning rate $l_\tau$ | 1e-3 |
|  | $V_\theta$ learning rate $l_\theta$ | 3e-4 |
|  | Policy learning rate $l_w$ | 3e-4 |
|  | Policy learning rate schedule | cosine annealing |
|  | $V_\theta$ gradient L2-regularization | 1e-4 |
|  | Discount factor | 0.99 |
|  | $f$-divergence | $\chi^2$ for Robomimic tasks
KL for other tasks |
|  | $\alpha$ | 4 for walker2d-medium-replay
0.5 for other D4RL tasks
0.5 for Antmaze tasks
2 for Lift and Can tasks
0.3 for Quadruped-walk + Quadruped-jump and
3 for the others in multi-task data sharing experiments |

The pseudocode of RGM with deep neural networks can be found in Algorithm 1. We run RGM on one RTX 3080Ti GPU with about 1h30min training time to apply 1M gradient steps.

We report the wall-clock training time of RGM compared with SOTA offline RL methods as well as SOTA offline IL methods that can learn from mixed quality data in Table 4. RGM is as efficient as most baselines but has an additional ability to combat the negative impacts of imperfect rewards.

Table 4: Wall-clock run time comparison of RGM and other baselines

| BC | DWBC | SMODICE | TD3+BC | CQL | IQL | RGM(ours) |
|---|---|---|---|---|---|---|
| 30min | 2h40min | 2h20min | 45min | 4h30min | 1h30min | 2h30min |

---

**Algorithm 1** RGM (KL-divergence) with Deep Neural Networks

---

**Input:** One Expert demonstration $\mathcal{D}^E$, offline Dataset $\mathcal{D}$, set $\mathcal{D} \leftarrow \mathcal{D}^E \cup \mathcal{D}$. Initialize $\tau, \phi, \theta, w$.
//​ Discriminator learning
Train $h_\tau$ using $\mathcal{D}^E$ and $\mathcal{D}$ using Eq. (14).
**for** $t = 0, 1, 2, ..., N$ **do**
    Sample mini-batch transitions $(s, a, \tilde{r}, s') \sim \mathcal{D}$
    //​ Reward Gap Minimization Bi-level optimization
    Update $V_\theta, \Delta r_\phi$ using Eq. (44) with $l_\phi \ll l_\theta$
    //​ Policy extraction
    Update $\pi_w$ based on Eq. (16) and Eq. (46)
**end for**

---

## C EXPERIMENTAL DETAILS

In this section, we introduce the detailed experimental setups in our paper.

### C.1 D4RL EXPERIMENTS

**Task Descriptions.** The D4RL (Fu et al., 2020) tasks we try to solve include Hopper, Halfcheetah and Walker2d. For these tasks, RL policies need to control the robots to move in the forward (right) direction by applying torques on the joints.

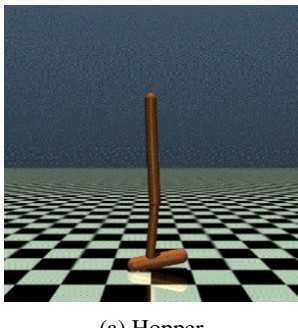
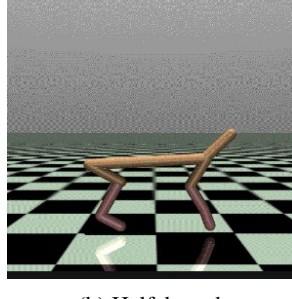
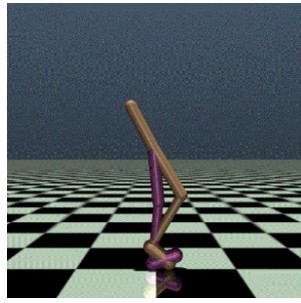

(a) Hopper        (b) Halfcheetah        (c) Walker2d

Figure 7: D4RL MuJoCo tasks

**Dataset composition.** The D4RL (Fu et al., 2020) datasets that we used in this paper contain 5 types of datasets: random: roll out a random policy for 1M steps. expert: roll out an expert policy that trained with SAC (Haarnoja et al., 2018) for 1M steps. medium: roll out a medium policy that achieves 1/3 the performance of the expert for 1M steps. medium-replay: replay buffer of a SAC agent that is trained to the performance of the medium policy. medium-expert: equally mixed dataset combines medium and expert data. We sample **only one trajectory from the expert dataset** to serve as the expert demonstration $\mathcal{D}^E$. The other datasets are treated as non-expert datasets $\mathcal{D}$.

Table 5: Dataset compositions for D4RL Experiments

| Task | State Dim | Expert Dataset | Number of Trajectories | Expert Data Size |
|------|-----------|----------------|------------------------|------------------|
| Hopper | 11 | hopper-expert-v2 | 1 | 1000 |
| Halfcheetah | 17 | halfcheetah-expert-v2 | 1 | 1000 |
| Walker2d | 17 | walker2d-expert-v2 | 1 | 1000 |

**Imperfect rewards.** We assume the original rewards in D4RL datasets are perfect, since we evaluate the policy performance based on the perfect reward function in the original gym environment during evaluation. We randomly flip the sign of $50\%$ original rewards to construct partially correct rewards, where half rewards can provide correct learning signals while the other half cannot. We flip all signs of the original rewards to construct completely incorrect rewards.

## C.2 SPARSE REWARD EXPERIMENTS

**Task descriptions.** The Robomimic (Mandlekar et al., 2021) tasks we try to solve include Lift and Can. For the Lift task, RL policy needs to control a 7-DOF robot arm to learn to lift a cube that is randomly located at a table. For the Can task, RL policy needs to control a 7-DOF robot arm to learn to pick a can that is randomly located at a table and place it in a specific location.

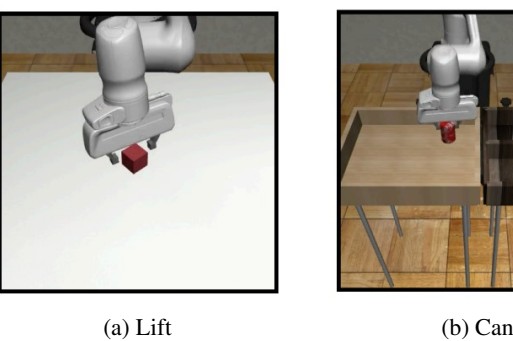

(a) Lift            (b) Can

Figure 8: Robomimic tasks

The AntMaze tasks we try to solve include AntMaze medium tasks, where an ant not only needs to learn to walk but also navigates from the goal to the destination in a medium-size maze. This task is extremely difficult due to the non-markovian and mixed-quality offline dataset, the stochastic property of environments, and the high dimensional state-action space (Fu et al., 2020).

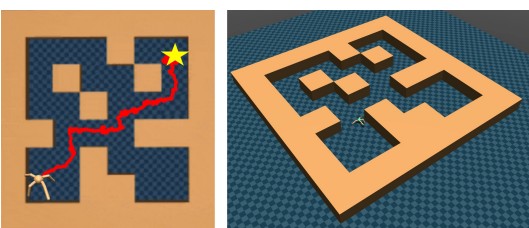

Figure 9: AntMaze medium task.

**Robomimic dataset composition.** The Robomimic (Mandlekar et al., 2021) datasets that we used in this paper contain 3 types of datasets: PH (Proficient-Human): datasets are collected by a single, experienced human operator. MH (Multi-Human): datasets are collected by 6 human operators of varying proficiency. MG (Machine-Generated): datasets are collected by first training SAC on the Lift and Can task, taking agent checkpoints that are saved regularly during training, and collecting 300 rollout trajectories from each checkpoint. We treat PH dataset as the expert dataset since the environment is stochastic, thus only one expert trajectory is difficult to capture the expert distribution. We use MG datasets as the large potentially suboptimal dataset rather than MH datasets since MH datasets are non-markovian and thus are hard to be solved by modern offline RL methods (Mandlekar et al., 2021), which is not the main challenge we try to solve.

Table 6: Dataset compositions for Robomimic Datasets

| Task | State Dim | Expert Dataset | Expert Size | Non-expert Dataset | Non expert Size |
|------|-----------|----------------|-------------|--------------------|-----------------|
| Lift | 19 | Lift-PH | 9666 | Lift-MG | 225K |
| Can | 23 | Can-PH | 23207 | Can-MG | 585K |

**AntMaze dataset composition.** The expert dataset of RGM is composed of 30 successful trajectories (which may be suboptimal) that are collected by training IQL with dense rewards. We set the original D4RL Antmaze-medium-play-v2 and Antmaze-medium-diverse-v2 datasets as non-expert datasets.

## C.3 MULTI-TASK DATA SHARING EXPERIMENTS

**Task descriptions.** The multi-task data sharing experiments contain 2 domains with 4 tasks per domain built on DeepMind Control Suite (Tassa et al., 2018). The immediate rewards in the 8 tasks are all in the unit interval, $r(s, a) \in [0, 1]$. (a) For **Walker** (*Stand, Walk, Run, Flip*) domain, the agent needs to control a biped in a 2D vertical plane to master four different locomotion skills. The observation space is 24 dimensional, and the action space is 6 dimensional. The episode length is set to 1000. (b) For **Quadruped** (*Walk, Run, Roll-Fast, Jump*) domain, the agent needs to control a quadruped within a 3D space to master four different moving skills. The observation space is 78 dimensional, and the action space is 12 dimensional. The episode length is set to 1000.

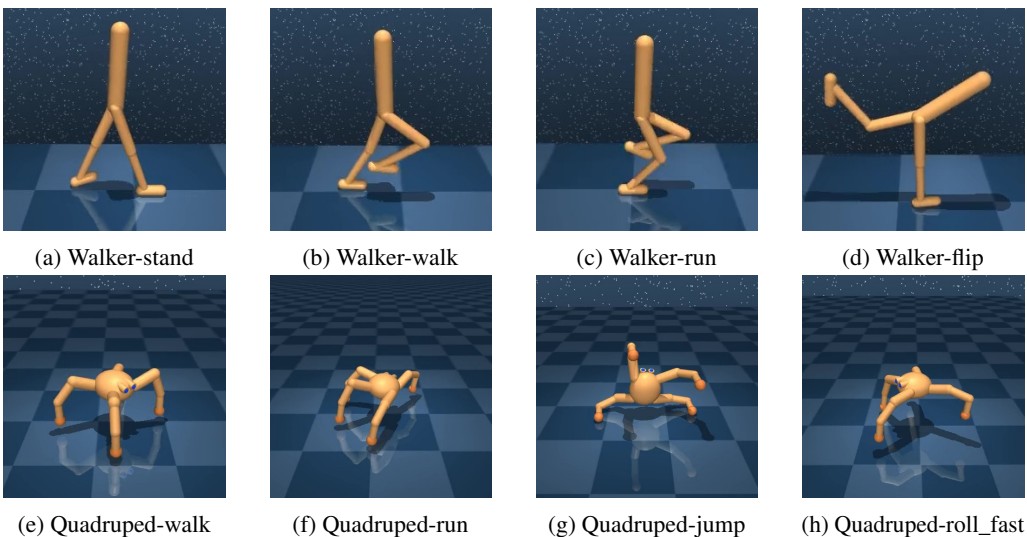

| (a) Walker-stand | (b) Walker-walk | (c) Walker-run | (d) Walker-flip |
| (e) Quadruped-walk | (f) Quadruped-run | (g) Quadruped-jump | (h) Quadruped-roll_fast |

Figure 10: Different tasks in Walker and Quadruped domain

**Dataset composition.** We take the same rule of dataset generation and similar task settings as the work (Bai et al., 2023). For each task, we utilize TD3 (Fujimoto et al., 2018) to collect three types of datasets (*expert, medium, replay*). The *expert* dataset contains only one expert episode, the *medium* dataset contains 1000 episodes of interactions, and the *replay* dataset contains 2000 episodes of interactions. For **Walker** (*Stand, Walk, Run, Flip*) domain, the *Stand* task is set to the target task, and the others are relevant tasks. For **Quadruped** (*Walk, Run, Jump, Roll-Fast*) domain, the *Walk* task is set to the target task, and the others are relevant tasks. We conduct two-task data sharing experiments, in which we share the *replay* dataset of the relevant task with the *medium* dataset of the target task.

## C.4 GRID WORLD EXPERIMENTS

**Dataset composition.** The offline dataset $\mathcal{D}$ we use in grid world experiments consists of 1000 trajectories generated by a completely random policy (Figure 11 (b)). There are two settings of imperfect rewards $\tilde{r}$: (i) $\tilde{r} = +10$ when reaching the goal while $\tilde{r} = 0$ anywhere else. (ii) (Figure 11 (c)) $\tilde{r} = +10$ when reaching the goal, $\tilde{r} = -10$ when encountering the fire (true fire or fake fire), $\tilde{r} = 0$ everywhere else. The expert demonstration dataset $\mathcal{D}^E$ consists of **only one expert demonstration** (Figure 11 (a)).

# D ADDITIONAL RESULTS

In this section, we provide additional comparative and ablation results of RGM against baseline methods.

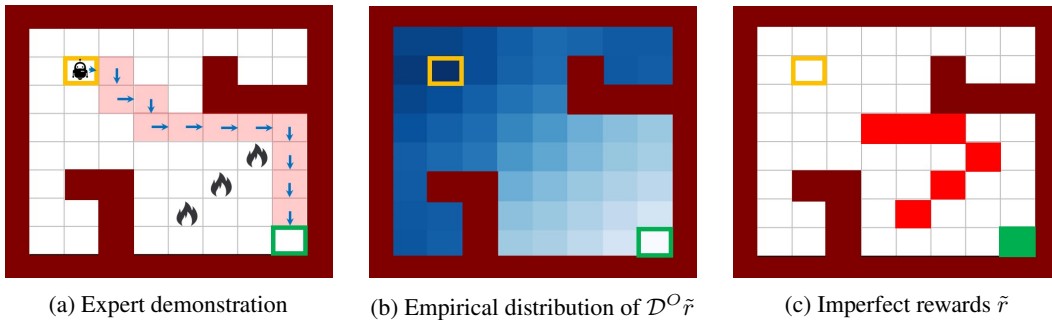

(a) Expert demonstration     (b) Empirical distribution of $\mathcal{D}^O\tilde{r}$     (c) Imperfect rewards $\tilde{r}$

Figure 11: (a) The only one expert demonstration path, which starts from ☐, follows the path ▨ and arrow ➡ to reach the goal ☐. (b) The empirical distribution heatmap of offline dataset $\mathcal{D}^O$, which consists of trajectories generated by random policy starting from ☐. The darker the color is, the more frequently the agent passes. (c) Illustration of imperfect rewards. Agent gets $\tilde{r} = -10$ when reaching ■, $\tilde{r} = +10$ when reaching ■, $\tilde{r} = 0$ everywhere else.

## D.1  ADDITIONAL COMPARISON TO OFFLINE IL

Recall that DWBC (Xu et al., 2022b) and SMODICE (Ma et al., 2022) all assume the offline dataset already covers a lot of expert trajectories, which is more restrictive compared to the requirement of RGM. Therefore, we further demonstrate the superiority of RGM compared to these offline IL methods by evaluating RGM under the same settings of DWBC and SMODICE. We combine the original D4RL dataset with 200 or 100 expert trajectories as the offline dataset $\mathcal{D}$, see Table 7 for descriptions of the expert trajectories. The comparisons under these dataset configurations can be found in Table 8. We can observe from Table 8 that RGM still outperforms existing SOTA offline IL methods under their settings.

Table 7: The details about the expert data that are used to construct the non-expert dataset in offline IL settings.

| Task | State Dim | Expert Dataset | Number of Trajectories | Expert Data Size |
|---|---|---|---|---|
| Hopper | 11 | hopper-expert-v2 | 200 | 193430 |
| Halfcheetah | 17 | halfcheetah-expert-v2 | 200 | 199800 |
| Walker2d | 17 | walker2d-expert-v2 | 100 | 99900 |

## D.2  EXPERIMENTS ON SAMPLING FROM DISCOUNTED DISTRIBUTIONS

We also implemented the discounted visitation distribution sampling in RGM. This is done by augmenting the D4RL datasets that adds the timestep of each $(s, a)$ pair in an episode. When performing sampling in Eq.(14-16) and calculating the gradient, we sample $(s, a, t)$ in the D4RL datasets and then multiply the gradient by $\gamma^t$. Empirically, we found that the performance of the discounted visitation distribution version is not better than the sampling distribution version of RGM. Figure 12 and Table 9 show that RGM (sampling distribution) surpasses RGM (discounted visitation distribution) in most cases with lower variance, while the latter wins by a slight margin in only a few cases.

## D.3  EXPERIMENTS ON NOISY PARTIALLY CORRECT REWARDS

We add i.i.d Gaussian noises with different standard deviation $\sigma$ to original D4RL rewards to construct noisy imperfect rewards with different degrees of imperfection. We set $\sigma = 1$ to construct partially correct rewards and $\sigma = 10$ as largely incorrect rewards, see Table 10 for detailed results.

Table 10 shows that RGM under perfect rewards slightly outperforms RGM with partially correct rewards, indicating that RGM can largely remedy the negative impacts caused by reward noises with $\sigma = 1$. Meanwhile, the highly noisy rewards ($\sigma = 10$) surely impact the performance, but its mean

Table 8: Average normalized scores of RGM compared with SOTA offline IL methods that can learn from mixed quality data under their settings. The notation "-w.e" stands for the mixed dataset that combines the original D4RL dataset with some expert trajectories. The scores are taken over the final 10 evaluations with 5 random seeds. We obtain the results via ruining author-provided open-source codes. RGM achieves 7 highest scores in 12 tasks.

| Dataset | BC | DWBC | SMODICE | RGM (Ours) |
|---|---|---|---|---|
| hopper-r-w.e | 2.8 | 59.5±30.3 | 108.7±5.4 | 110.4 ±1.2 |
| halfcheetah-r-w.e | 0.2 | 3.3±1.7 | 89.3 ±1.5 | 57.6±6.4 |
| walker2d-r-w.e | 1.2 | 81.8±0.6 | 102.0±9.9 | 109.2 ±0.2 |
| hopper-m-w.e | 54.9 | 39.0±22.5 | 54.5±4.3 | 66.1 ±9.7 |
| halfcheetah-m-w.e | 41.2 | 8.5±9.2 | 55.4 ±10.9 | 50.5±7.9 |
| walker2d-m-w.e | 62.1 | 56.1±39.2 | 6.5±9.7 | 79.2 ±12.4 |
| hopper-m-r-w.e | 23.4 | 23.1±17.1 | 53.0±27.5 | 58.6 ±27.0 |
| halfcheetah-m-r-w.e | 24.2 | 1.1±1.2 | 84.9 ±7.2 | 65.4±15.1 |
| walker2d-m-r-w.e | 21.8 | 85.5 ±33.6 | 8.9±12.3 | 71.7±32.6 |
| hopper-m-e-w.e | 51.2 | 40.0±22.5 | 71.4±15.5 | 89.1 ±13.5 |
| halfcheetah-m-e-w.e | 61.7 | 1.2±0.5 | 87.2 ±1.9 | 76.8±8.8 |
| walker2d-m-e-w.e | 103.2 | 76.8±31.7 | 14.1±2.0 | 105.8 ±8.6 |
| Mean Score | 37.8 | 39.7±17.5 | 61.3±9.0 | 78.3 ±12.0 |

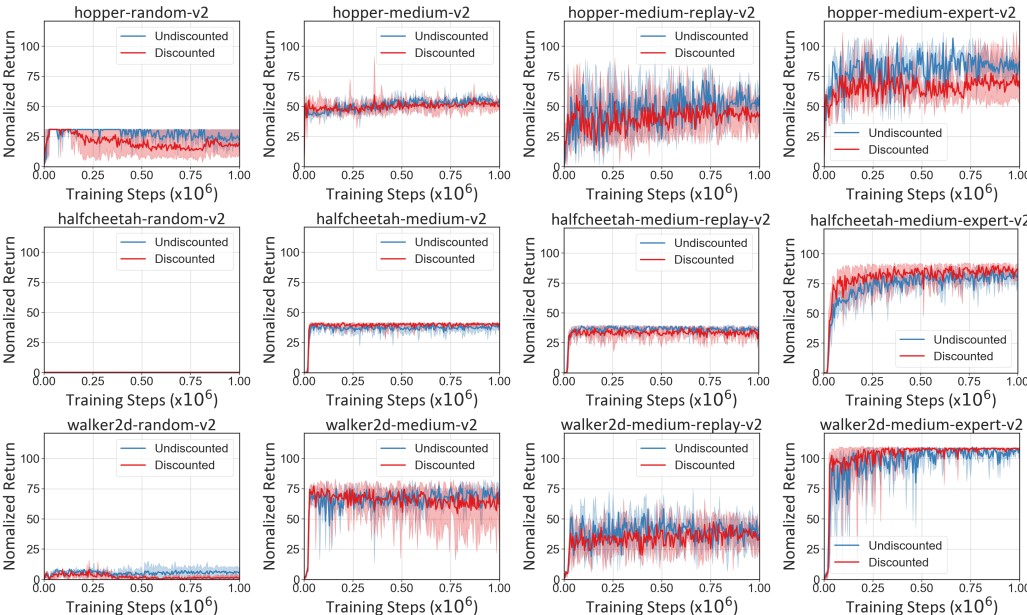

Figure 12: Experiments on sampling from discounted and undiscounted distributions

score is 45.0, which is still considerably higher than other Offline RL and IL methods under partially correct rewards with the largest mean value of 35.5 as shown in Table 1.

## D.4    ABLATIONS ON THE NUMBER OF EXPERT TRAJECTORIES

We add the ablations on the number of expert trajectories in $\mathcal{D}^E$ ($N^E$) for RGM, SMODICE and DWBC. Table 11, 12 and 13 show that RGM achieves better performance than offline IL methods

Table 9: Normalized scores of RGM sampling from discounted distribution and undiscounted distribution

| Dataset | RGM (Discounted) | RGM (Undiscounted) |
|---|---|---|
| hopper-r | 19.8±0.2 | 21.2 ±0.4 |
| halfcheetah-r | 0.2 ±0.0 | 0.2 ±0.0 |
| walker2d-r | 1.2±1.7 | 7.7 ±3.3 |
| hopper-m | 51.1±4.9 | 55.5 ±1.0 |
| halfcheetah-m | 40.3±1.6 | 40.7 ±1.4 |
| walker2d-m | 62.2±22.5 | 72.3 ±10.7 |
| hopper-m-r | 43.3±11.6 | 59.1 ±15.3 |
| halfcheetah-m-r | 34.5±4.5 | 37.8 ±2.6 |
| walker2d-m-r | 34.3±11.0 | 48.6 ±3.6 |
| hopper-m-e | 65.3±19.5 | 87.1 ±10.7 |
| halfcheetah-m-e | 87.3 ±7.8 | 81.5±0.8 |
| walker2d-m-e | 108.4±0.6 | 108.8 ±0.4 |
| Mean score | 45.7±7.2 | 52.0 ±4.2 |

Table 10: Normalized scores of RGM on different degrees of noisy datasets.

| Dataset | RGM(T) | RGM ($\sigma = 1$) | RGM ($\sigma = 10$) |
|---|---|---|---|
| hopper-r | 29.6 | 8.5 | 9.8 |
| halfcheetah-r | 0.2 | 0.3 | 0.2 |
| walker2d-r | 3.9 | 0.6 | -0.1 |
| hopper-m | 56.2 | 52.0 | 47.9 |
| halfcheetah-m | 40.4 | 41.2 | 38.4 |
| walker2d-m | 73.3 | 71.9 | 72 |
| hopper-m-r | 60.3 | 58.0 | 40.0 |
| halfcheetah-m-r | 37.9 | 38.3 | 28.1 |
| walker2d-m-r | 46.3 | 42.5 | 43.8 |
| hopper-m-e | 106.1 | 82.0 | 82.8 |
| halfcheetah-m-e | 85.6 | 88.7 | 69.1 |
| walker2d-m-e | 109.2 | 108.2 | 108.5 |
| Mean score | 54.1 | 49.4 | 45.0 |

designed for mixed-quality data (DWBC and SMODICE). It is found that RGM also enjoys a higher level of performance gains when the amount of expert data is increased.

## D.5 Experiments on multi-task data sharing

We present concrete results of the multi-task data sharing experiment. Table 14 shows the evaluated scores on multi-task data sharing, which are illustrated in Fig. 3.

## D.6 Additional learning curves of RGM

We present the learning curves of RGM compared with offline IL and RL baselines on D4RL datasets related to the results presented in Table 1.

## D.7 Illustrative example for the non-tabular scenarios

The results of the 8×8 grid world experiments in Section 5.2 and Appendix C.4 illustrate the potential benefits of the learned rewards in the tabular case. In this subsection, we consider a one-dimensional

Table 11: Normalized scores of RGM and offline IL baselines when $\mathcal{D}^E$ contains 10 expert trajectories.

| Dataset | DWBC ($N^E = 10$) | SMODICE ($N^E = 10$) | RGM ($N^E = 10$) |
|---|---|---|---|
| hopper-r | 52.5 | 1.3 | 30.8 |
| halfcheetah-r | -0.3 | 2.1 | 0.2 |
| walker2d-r | 96.2 | 0.3 | 6.1 |
| hopper-m | 31.1 | 53.8 | 54.5 |
| halfcheetah-m | 5.0 | 40.9 | 41.4 |
| walker2d-m | 22.4 | 3.3 | 72.9 |
| hopper-m-r | 37.4 | 33.2 | 55.5 |
| halfcheetah-m-r | 3.9 | 36.7 | 34.9 |
| walker2d-m-r | 90.7 | 34.7 | 43.1 |
| hopper-m-e | 31.2 | 85.0 | 89.2 |
| halfcheetah-m-e | 10.9 | 86.6 | 79.4 |
| walker2d-m-e | 46.3 | 14.1 | 109.0 |
| Mean score | 35.6 | 32.7 | 51.4 |

Table 12: Normalized scores of RGM and offline IL baselines when $\mathcal{D}^E$ contains 40 expert trajectories.

| Dataset | DWBC ($N^E = 40$) | SMODICE ($N^E = 40$) | RGM ($N^E = 40$) |
|---|---|---|---|
| hopper-r | 54.6 | 67.2 | 36.9 |
| halfcheetah-r | 8.8 | 14.8 | 18.7 |
| walker2d-r | 78.7 | 92.9 | -0.1 |
| hopper-m | 13.5 | 54.2 | 57.0 |
| halfcheetah-m | 5.6 | 44.5 | 40.9 |
| walker2d-m | 16.9 | 3.5 | 74.3 |
| hopper-m-r | 54.3 | 47.2 | 54.3 |
| halfcheetah-m-r | 46.1 | 54.6 | 46.1 |
| walker2d-m-r | 87.8 | 35.7 | 61.2 |
| hopper-m-e | 34.5 | 75.9 | 92.4 |
| halfcheetah-m-e | 3.4 | 85.1 | 84.7 |
| walker2d-m-e | 57.2 | 17.1 | 108.6 |
| Mean score | 38.5 | 44.9 | 56.5 |

random walk task in the non-tabular case and provide the visualization of the learned corrected rewards $\hat{r}$. In this task, the state space is a straight line from [0, +3] and the agent can move at each step in the range of [-0.5, 0.5]. If the agent goes beyond the edge ($s < 0$ or $s > +3$), then we keep it at the edge ($s = 0$ or $s = +3$). The agent needs to start from state $s = 0$ and reach the destination located at $s = 3$ as fast as possible. The expert dataset $\mathcal{D}^E$ consists of one trajectory where the expert takes action $a = 0.5$ at every state. The offline dataset $\mathcal{D}$ consists of 1000 trajectories generated by a completely random policy where the agent takes action uniformly from [-0.5, 0.5] at every state. The sparse rewards $\tilde{r} = +10$ is set when reaching the destination while $\tilde{r} = 0$ anywhere else. The visualization of learned rewards $\hat{r}$ at each state-action pair is shown in Figure 14.

# E    DISCUSSION ON THE APPLICABILITY TO ONLINE SETTINGS

It should be noted that the proposed RGM framework can also be applied to the online setting. This can be achieved by simply setting $\alpha = 0$ in Eq. (4-5), and we have the bi-level objective of the online

Table 13: Normalized scores of RGM and offline IL baselines when $\mathcal{D}^E$ contains 80 expert trajectories.

| Dataset | DWBC ($N^E = 80$) | SMODICE ($N^E = 80$) | RGM ($N^E = 80$) |
|---|---|---|---|
| hopper-r | 65.1 | 92.7 | 47.3 |
| halfcheetah-r | 2.3 | 48.1 | 40.4 |
| walker2d-r | 86.1 | 98.8 | 109.1 |
| hopper-m | 8.8 | 53.4 | 59.7 |
| halfcheetah-m | 6.7 | 51.2 | 42.5 |
| walker2d-m | 36.5 | 2.9 | 73.4 |
| hopper-m-r | 35.3 | 46.6 | 66.2 |
| halfcheetah-m-r | 36.1 | 59.6 | 54.0 |
| walker2d-m-r | 85.8 | 32.8 | 64.9 |
| hopper-m-e | 8.8 | 83.7 | 97.1 |
| halfcheetah-m-e | 12.1 | 87.4 | 83.9 |
| walker2d-m-e | 64.5 | 43.7 | 108.6 |
| Mean score | 37.5 | 58.4 | 70.6 |

Table 14: Evaluated scores on multi-task data sharing.

| Domain | Dataset | CDS | CDS+UDS | RGM |
|---|---|---|---|---|
| Walker | stand-medium + walk-replay | 486.1±7.2 | 415.3±44.8 | 753.3±107.6 |
| Walker | stand-medium + run-replay | 455.8±21.9 | 440.0±8.4 | 620.0±31.0 |
| Walker | stand-medium + flip-replay | 492.4±19.0 | 371.2±123.0 | 745.6±100.8 |
| Quadruped | walk-medium + run-replay | 527.6±213.0 | 155.7±53.0 | 900.0±48.6 |
| Quadruped | walk-medium + roll_fast-replay | 476.2±45.1 | 439.1±90.7 | 493.2±37.3 |
| Quadruped | walk-medium + jump-replay | 533.5±168.5 | 521.9±236.9 | 490.9±119.4 |
| Mean score | | 495.3±79.1 | 390.5±92.8 | 667.0±74.1 |

version of RGM:

$$\Delta r^* = \arg\min_{\Delta r} \; D_f \left( d^{\pi_{\hat{r}}^*} \| d^E \right)$$
$$\text{s.t.} \quad \pi_{\hat{r}}^* = \arg\max_{\pi} \; \mathbb{E}_{(s,a) \sim d^{\pi_{\hat{r}}}} [\hat{r}(s,a)] \tag{49}$$

Since we could get online samples from $d^{\pi_{\hat{r}}^*}$ in the online setting, so we don't have to eliminate $d^{\pi_{\hat{r}}^*}$. One can use the existing popular online RL algorithms to solve the lower-level problem, while leveraging the online samples from $d^{\pi_{\hat{r}}^*}$ to solve the upper-level problem. Hence the online version of RGM can be perceived as a reduced and simplified version of the original RGM. The core idea of the reward correction has not been changed in the online setting, which illustrates that to some extent, our proposed RGM is a unified policy optimization method for imperfect rewards.

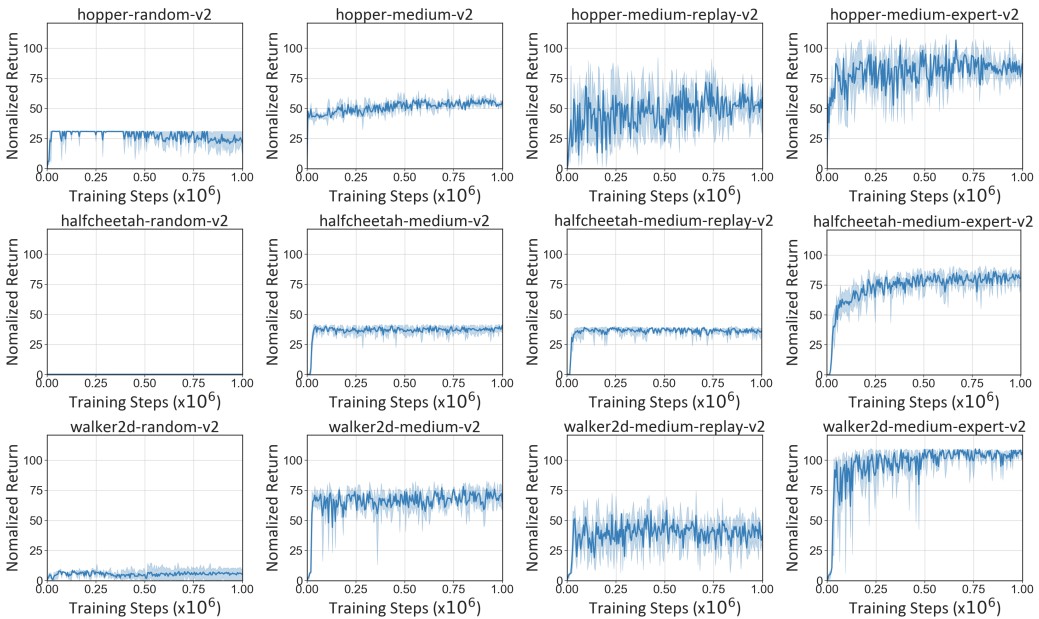

Figure 13: Learning curves of RGM trained on D4RL datasets under imperfect rewards.

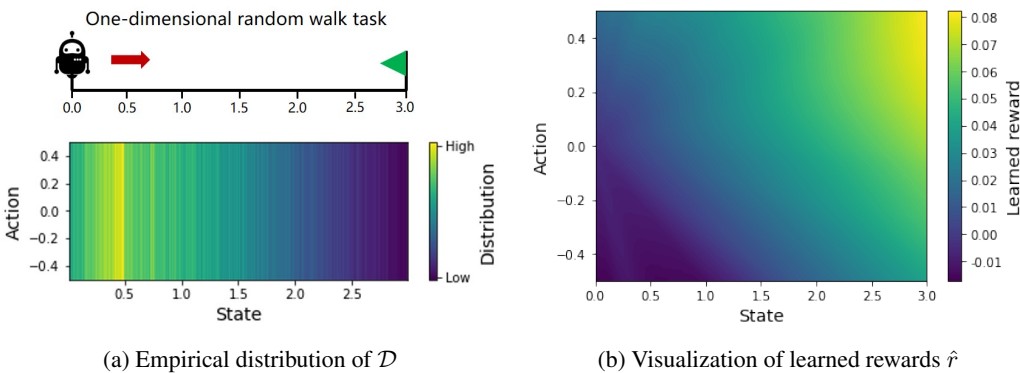

(a) Empirical distribution of $\mathcal{D}$       (b) Visualization of learned rewards $\hat{r}$

Figure 14: (a) The empirical distribution of offline dataset $\mathcal{D}$ in a continuous one-dimensional random walk task. Most states in the offline dataset are distributed near the starting point. (b) At each state (at each vertical line), the learned reward $\hat{r}$ gets a larger value when the action gets closer to 0.5. The expert data has contain 7 states ($s = 0.0, 0.5, 1.0, 1.5, 2.0, 2.5, 3.0$), but the learned rewards can still generalize well in the state space even in regions that are not covered by the expert data. Similar to the 8×8 grid world experiment, we can successfully navigate to the destination by only maximizing per-step reward $\hat{r}$, which means that the learned rewards also encode long-horizon information.

