# OpenReview forum: "Mind the Gap: Offline Policy Optimization for Imperfect Rewards"
_ICLR.cc/2023/Conference — ICLR 2023 poster_

### Official Review · Reviewer_gH6T · 2022-10-23

**Confidence:** 3
**Correctness:** 3
**Technical Novelty And Significance:** 3
**Empirical Novelty And Significance:** 3
**Recommendation:** 8

**Clarity, Quality, Novelty And Reproducibility:**

The description of the problem, of the approach, and of the experiments is generally really clear.

Things to improve:
- the last paragraph about ablations + Fig 5 could be made a bit clearer, I found it hard to parse.
- The r/m/m-r/m-e versions of the datasets in Table 1 should be explained in the text (maybe it’s there but couldn’t find it, had to look in the D4RL paper)


**Strength And Weaknesses:**

The right context and background is provided to describe the approach. The proposed RGM formulation is intriguing and seems natural mathematically, even if intuitively it seems like a challenging optimization objective.

As far as I could tell, all derivations are correct and they are explained clearly. The experimental section is generally well designed, with some good illustrative examples in Fig 3 and a number of useful experiments to establish the validity of the method.

Some questions and comments:

- The classification problem for h seems hard in general. As if I understand correctly, it amounts to detecting whether an action is optimal or not. Couldn’t this by itself, if learned perfectly, be used to generate the optimal/expert policy.
- It would be useful to have some notion of the computational cost of the method (e.g. as a function of the dataset size) and how that compares to the perfect reward case.
- It would be nice to visualize how well the reward is recovered in the non-tabular scenarios.
- The fact that RGM does better than other offline RL methods even with perfect rewards seems a bit surprising (sec 5.1). Why should this be expected?
- It seems a bit unfair to evaluate offline IL approaches that expect expert data on these mixed datasets. Are there other methods that can deal naturally with the B (imperfect reward) condition?
- The RGM formulation makes no assumption on the way the rewards might be corrupted (versus for example some noise process etc.) It would be good to comment on the pros/cons of that.
- Could you clarify how the approach relates to "Generative Adversarial Imitation Learning"?


**Summary Of The Paper:**

The authors study the offline RL setting with imperfect rewards. That is, a known portion of the trajectories contain possibly corrupted rewards. The approach proposed to tackle this problem is Reward Gap Minimization (RGM), which finds the reward function (correction) which, when used to solve for the optimal policy, agrees with the expert trajectory distribution. This requires a dual-optimization problem to be solved and a number of reformulations are provided to get a practical form. The approach is empirically tested, with some analysis and ablations, in three continuous control problems (hopper/half-cheetah/hopper).

**Summary Of The Review:**

Good and clear paper about a principled and practical way to deal with imperfect rewards in offline RL. I think the paper has enough empirical evidence as it is, but I do wonder whether the approach can be scaled to more complex settings (larger datasets, more difficult/higher-dim task).

---

> ### Author Response · Authors · 2022-11-12
> **Reference**
>
> [1] Kostrikov, Ilya, et al. Offline reinforcement learning with fisher divergence critic regularization. *International Conference on Machine Learning*. PMLR, 2021.
>
> [2] Kumar, Aviral, et al. Conservative q-learning for offline reinforcement learning. *Advances in Neural Information Processing Systems* 33 (2020): 1179-1191.
>
> [3] Sun, Hao, et al. Exploiting reward shifting in value-based deep rl. *arXiv preprint arXiv:2209.07288* (2022).
>
> [4] David Abel, et al. On the expressivity of markov reward. *Advances in Neural Information Processing Systems, 34:7799–7812, 2021*

---

> ### Author Response · Authors · 2022-11-12
> **Response to Reviewer gH6T W5~W7**
>
> > **W5. It seems a bit unfair to evaluate offline IL approaches that expect expert data on these mixed datasets. Are there other methods that can deal naturally with the B (imperfect reward) condition?**
>
> - The existing offline IL methods heavily depend on the size and quality of expert data to learn good policies, otherwise will suffer from severe covariate shifts and compounding errors. There are some recent offline IL approaches, such as DWBC (Xu et al., 2022b) and SMODICE (Ma et al., 2022) alleviate this issue by learning from a relatively small expert dataset and a large suboptimal dataset, which are similar to the input setting of RGM. However, these methods still require a reasonable amount of expert data and are not able to incorporate additional reward information. In our experiments, we have compared RGM against DWBC and SMODICE, and show the superior performance of RGM (please refer to Section 5.2 for details).
> - The existing offline RL methods are not specifically designed to handle imperfect rewards. They suffer from severe performance drops under imperfect reward signals. Reward shaping methods can handle imperfect rewards, but there is no reward shaping or correction mechanism existing in the offline setting. Therefore, to our best knowledge, there is no existing method that can simultaneously enable learning from small expert data as well as imperfect reward condition.
>
> > **W6. The RGM formulation makes no assumption on the way the rewards might be corrupted (versus for example some noise process etc.) It would be good to comment on the pros/cons of that.**
>
> - We thank the reviewer for this comment. RGM is designed to handle diverse settings of rewards in offline policy optimization, so we choose not to impose extra assumptions or restrictions on the quality of the given rewards $\tilde{r}$. In our revised paper, we add additional experiments with different types of imperfect rewards, including completely wrong rewards, noisy rewards with Gaussian noises, and sparse rewards. Please refer to the response for W6 of reviewer 1KgB and W4, W5 of reviewer kWzN for detailed discussion.
>
> > **W7. Could you clarify how the approach relates to "Generative Adversarial Imitation Learning"?**
>
> - We compare our approach RGM to GAIL and provide the main differences and similarities between the two approaches as follows:
>   - **Differences**. First, as we mentioned in the response to the first comment, the discriminator training (Eq.(16)) in RGM is not through a GAN-like training paradigm. It does not need learning a generator but simply learning a discriminative classifier through supervised learning. In contrast, GAIL follows a similar training paradigm as GAN, since it updates the policy $\pi_{\theta}$ as the generative model to confuse a discriminative classifier $D_w(s,a)$. Second, GAIL cannot be used in the offline setting, due to its intrinsic online and on-policy optimization scheme; while RGM can be applied in both offline and online (see discussion in our response to W5 of reviewer 1KgB) setting. Finally, GAIL is essentially a state-action distribution matching method and is not specifically designed to handle imperfect rewards (the reward function for on-policy RL training is provided by a discriminator), while RGM solves the bi-level problem to handle diverse reward settings.
>   - **Similarities**. GAIL essentially minimizes the Jensen-Shannon divergence between $d^{\pi}$ and $d^{E}$. Similarly, RGM also minimizes the $f$-divergence between $d^{\pi^*_{\hat{r}}}$ and $d^E$ in the upper level problem (Eq.(6)), with $d^{\pi^*_{\hat{r}}}$ induced by the optimal policy $\pi^{*}_{\hat{r}}$ from the lower level RL problem (Eq.(7)) related to the reward $\hat{r}$.
>
> >**The last paragraph about ablations + Fig 5 could be made a bit clearer, I found it hard to parse.**
> - We thank the reviewer for the suggestion! We have revised the corresponding part in our updated paper.
>
> >**The r/m/m-r/m-e versions of the datasets in Table 1 should be explained in the text (maybe it’s there but couldn’t find it, had to look in the D4RL paper)**
> - Thank the reviewer for the suggestion! We have added the explanation of the abbreviation in the caption of Table 1.

---

> ### Author Response · Authors · 2022-11-12
> **Response to Reviewer gH6T W1~W4**
>
> We sincerely appreciate the reviewer for the kind and constructive comments. Below are detailed responses to each comment:
>
> > **W1. The classification problem for h seems hard in general. As if I understand correctly, it amounts to detecting whether an action is optimal or not. Couldn’t this by itself, if learned perfectly, be used to generate the optimal/expert policy.**
>
> - The reviewer has raised an interesting and important point on the discriminator training (Eq.(16)). In fact, Eq.(16) is not strictly the training paradigm of GAN, because there is no generator training in Eq.(16). We can simply minimize Eq.(16) to get $d^{E}/d^{\mathcal{D}}$ before solving the bi-level problem. It is a simple minimization problem solved via supervised learning, instead of a minimax problem or generative adversarial problem. So in general, solving Eq.(16) is not too hard. If $h(s,a)$ is learned perfectly for every point in the state-action space, then $h(s,a)$ in principle could be used to generate the expert policy. However, since our method only requires a small set of expert data, learning a perfect $h(s,a)$ is generally not possible, especially in some states not covered by expert data. Hence using it as a form of distribution ratio rather than directly using it to generate optimal/expert policy serves a better purpose in our offline policy optimization problem.
>
> >**W2. It would be useful to have some notion of the computational cost of the method (e.g. as a function of the dataset size) and how that compares to the perfect reward case.**
> - We thank the reviewer for the comment. We provide the wall-clock run time of RGM and other baselines as follows. We also updated these results in Appendix B.3 of our revised paper. RGM is as efficient as most baselines but has an additional ability to combat the negative effect of imperfect rewards.
>
> |BC|DWBC|SMODICE|TD3+BC|CQL|IQL|RGM (ours)|
> |---|---|---|---|---|---|---|
> |30min|2h40min|2h20min|45min|4h30h|1h30min|2h30min|
>
> >**W3. It would be nice to visualize how well the reward is recovered in the non-tabular scenarios.**
>
> - Thank the reviewer for the suggestion! We add an illustrative example for the non-tabular scenarios in Appendix E.7 of our revised paper. Please check our revised paper for details.
>
> > **W4. The fact that RGM does better than other offline RL methods even with perfect rewards seems a bit surprising (sec 5.1). Why should this be expected?**
>
> - Actually, RGM achieves similar performance to offline RL policies that are trained on perfect rewards. In some cases such as hopper-r and walker2d-r, RGM can surpass offline RL with perfect rewards, while in other cases such as halfcheetah-r and hopper-m-r, offline RL with perfect rewards might be better. We provide some reasons why RGM can sometimes surpass offline RL methods.
>   - Even given the perfect rewards, some offline RL methods [1-2] still need to reshape or shift the rewards to achieve the best policy performance, the equivalence to engineering the initialization of Q-function estimation that encourages conservative exploitation under offline learning (see reference [3] below for a more in-depth discussion).
>   - As we stated in Section 5.3, the learned rewards can encode long horizon information. This can be more informative than the original perfect per-step rewards and is helpful for faster RL policy learning [4]. For example, in the demonstrative experiments in Figure 3, we can even successfully navigate to the destination at most positions by only maximizing the one-step reward.

---

> ### Author Response · Authors · 2022-11-22
> **Follow up**
>
> Dear reviewer gH6T,
>
> We sincerely appreciate your positive feedback on our paper! Your comments and suggestions will be of great help in improving the quality and clarity of our paper. We have added more experiments and clarifications in the latest revisions. We would really appreciate it if you could tell us if your concerns are resolved. If you have further concerns or questions, please tell us; they will greatly help to improve the quality of our paper and we will be more than happy to engage in further discussions and address them.
>
> Thanks!

---

> > ### Comment · Reviewer_gH6T · 2022-12-07
> > **Rebuttal response**
> >
> > Thank you for your answers and the clarifications, the appendix Fig E7 is a nice addition.

---

> > > ### Author Response · Authors · 2022-12-08
> > > **Thanks for the response!**
> > >
> > > Dear reviewer gH6T,
> > >
> > > Thank you for the response and your positive feedback on our paper! We really appreciate your time engaged in the review and discussion phase and we feel our work has been greatly improved through your constructive comments!
> > >
> > > Thanks!
> > >
> > > Best regards,
> > >
> > > Authors of Paper 448

---

### Official Review · Reviewer_KWzN · 2022-10-27

**Confidence:** 4
**Correctness:** 3
**Technical Novelty And Significance:** 3
**Empirical Novelty And Significance:** 2
**Recommendation:** 5

**Clarity, Quality, Novelty And Reproducibility:**

This paper is easy to follow. In addition, combining OptiDICE with reward correction term learning is interesting.

**Strength And Weaknesses:**

## **Strength**
- RGM uses bi-level optimization and this approach is very interesting.
## **Weakness**
I have following questions and concerns:
### **Theory**
- RGM does not consider the degree of imperfection of rewards. Theoretically, it seems that the degree has no effect on the performance of RGM - it is weird to me. Intuitively, if rewards are perfect, RGM should perform better than offline IL algorithms and comparable to offline RL algorithms, and if rewards are too imperfect, RGM should perform poorly. Consequently, it is questionable whether the imperfection of rewards affects RGM performance.
- (Related to the previous question) Suppose that we can obtain optimal values for each optimization problem. In this ideal case, RGM obtains the optimal reward correction term, optimal stationary distribution, and optimal policy. Then, what is the main difference between RGM and existing offline IL algorithms in theory? I think RGM is equivalent to DemoDICE (if f-divergence is KL divergence) and SMODICE if the max / min can be computed exactly.
- In footnote 4, page 8, the authors claim that “SMODICE and DemoDICE treat $\log\frac{d^E}{d^D}$ as rewards, which can only provide correct signals when $d^E>0$.“. However, in thes upper level problem (learning reward correction term), RGM also uses $w:=\frac{d^E}{d^D}$, especially in Eq. (17) – therefore, in my opinion, learnable reward correction term also can provide correct signals when $d^E>0$.
### **Experiments**
- In the beginning of the paper, the authors claim that the proposed methods can handle all three types of rewards (perfect, partially correct, incorrect). However, in the experiments consisting of D4RL dataset, there is only one imperfect reward setting (50% of rewards are sign-flipped). Thus, I think there should be additional empirical studies on (1) perfect and incorrect rewards (2) other types of imperfect rewards (e.g. adding i.i.d. gaussian noise for each reward).
- There should be RGM (T) in Table 1.
- What is your intuition about imperfect rewards used in the experiments in this paper? I do not think flipping the signs is a natural setting.
- In addition, I am curious about the ablation study on performance of RGM, offline RL algorithms, and offline IL algorithms according to (1) the degree of imperfection of rewards (including perfect rewards and completely incorrect reward) (2) the number of expert trajectories in $D^E$.


**Summary Of The Paper:**

This paper proposes a new unified offline policy optimization approach, named RGM (Reward gap minimization), that can handle imperfect rewards. RGM has two layers: (1) upper layer learns reward correction term by minimizing the reward gap towards expert behaviors, (2) lower layer solves a pessimistic RL problem under the corrected rewards. More precisely, the authors use f-divergence matching to learn the correction term, and OptiDICE to solve the pessimistic RL problem. Then, policy extraction is used to extract optimal policy from the learned stationary distribution. Finally, this paper provides an empirical study on D4RL datasets.

**Summary Of The Review:**

I think this paper proposes a novel algorithm to handle imperfect rewards. However, I have some questions and I believe they are important in evaluating this paper. Thus, I vote to weakly reject until these questions are resolved.

---

> ### Author Response · Authors · 2022-11-12
> **Response to Reviewer KWzN W5~W7**
>
> >**W5. There should be RGM (T) in Table 1.**
>
> - Thank you for the suggestions. We have add RGM (T) in Table 1. See responses to W4 for additional discussions.
>
> >**W6. What is your intuition about imperfect rewards used in the experiments in this paper? I do not think flipping the signs is a natural setting.**
>
> - Flipping the signs can straightforwardly get a **partially correct** imperfect reward function that some parts of rewards are perfect but others are wrong. This is different from simply adding i.i.d gaussian noise because a small gaussian noise may only make the rewards noisy instead of incorrect. Moreover, optimizing the policy based on the flipping signs is far more challenging, because flipping signs can produce completely contradicting learning signals on similar state-action pairs.
>
> >**W7. In addition, I am curious about the ablation study on performance of RGM, offline RL algorithms, and offline IL algorithms according to (1) the degree of imperfection of rewards (including perfect rewards and completely incorrect reward) (2) the number of expert trajectories in DE.**
>
> - We add experiments on offline RL algorithms and RGM trained on completely incorrect rewards (all reward signs are flipped). Such rewards are catastrophic for offline RL methods because it provides completely reversed learning signals for policy optimization.
>
> |Dataset|CQL|IQL|TD3+BC|RGM(ours)|
> |---|---|---|---|---|
> |hopper-r|0.0|0.7|0.4|**25.9**|
> |halfcheetah-r|-38.4|**2.2**|-11.6|0.2|
> |walker2d-r|-0.0|-0.3|**2.0**|-0.1|
> |hopper-m|11.2|35.5|37.3|**54.6**|
> |halfcheetah-m|4.1|35.4|1.2|**40.3**|
> |walker2d-m|3.3|22.0|17.2|**38.4**|
> |hopper-m-r|0.0|0.7|16.3|**44.5**|
> |halfcheetah-m-r|-1.1|1.8|-1.0|**29.8**|
> |walker2d-m-r|-0.0|-0.2|-0.7|**46.1**|
> |hopper-m-e|11.6|13.6|22.3|**66**|
> |halfcheetah-m-e|11.1|35.8|1.9|**78.4**|
> |walker2d-m-e|16.6|20.9|6.8|**108.8**|
> |Mean score|-0.3|14.0|7.7|**41.9**|
>
> - We also add the ablations on different numbers of expert trajectories in $\mathcal{D}_E$ for RGM, SMODICE and DWBC. The results show that RGM achieves better performance than offline IL methods that can learn from mixed-quality data (DWBC and SMODICE). It is also found that RGM also enjoys a higher level of performance gains when the amount of expert data is increased.
>
> ---
> |Dataset |DWBC ($N_E=10$) | SMODICE ($N_E=10$)| RGM ($N_E=10$)|
> |---|---|---|---|
> |hopper-r|**52.5**|1.3|30.8|
> |halfcheetah-r|-0.03|**2.1**|0.2|
> |walker2d-r|**96.2**|0.3|6.1|
> |hopper-m|31.1|53.8|**54.5**|
> |halfcheetah-m|5.0|40.9|**41.4**|
> |walker2d-m|22.4|3.3|**72.9**|
> |hopper-m-r|37.4|33.2|**55.5**|
> |halfcheetah-m-r|3.9|**36.7**|34.9|
> |walker2d-m-r|**90.7**|34.7|43.1|
> |hopper-m-e|31.2|85.0|**89.2**|
> |halfcheetah-m-e|10.9|**86.6**|79.4|
> |walker2d-m-e|46.3|14.1|**109.0**|
> |Mean score|35.6|32.7|**51.4**|
>
> |Dataset |DWBC ($N_E=40$) | SMODICE ($N_E=40$)| RGM ($N_E=40$)|
> |---|---|---|---|
> |hopper-r|54.6|**67.2**|36.9|
> |halfcheetah-r|8.8|14.8|**18.7**|
> |walker2d-r|78.7|**92.9**|-0.1|
> |hopper-m|13.5|54.2|**57.0**|
> |halfcheetah-m|5.6|**44.5**|40.9|
> |walker2d-m|16.9|3.5|**74.3**|
> |hopper-m-r|**54.3**|47.2|**54.3**|
> |halfcheetah-m-r|46.1|**54.6**|46.1|
> |walker2d-m-r|**87.8**|35.7|61.2|
> |hopper-m-e|34.5|75.9|**92.4**|
> |halfcheetah-m-e|3.4|**85.1**|84.7|
> |walker2d-m-e|57.2|17.1|**108.6**|
> |Mean score|38.5|44.9|**56.5**|
>
>
> |Dataset |DWBC ($N_E=80$) | SMODICE ($N_E=80$)| RGM ($N_E=80$)|
> |---|---|---|---|
> |hopper-r|65.1|**92.7**|47.3|
> |halfcheetah-r|2.3|**48.1**|40.4|
> |walker2d-r|86.1|98.8|**109.1**|
> |hopper-m|8.8|53.4|**59.7**|
> |halfcheetah-m|6.7|**51.2**|42.5|
> |walker2d-m|36.5|2.9|**73.4**|
> |hopper-m-r|35.3|46.6|**66.2**|
> |halfcheetah-m-r|36.1|**59.6**|54.0|
> |walker2d-m-r|**85.8**|32.8|64.9|
> |hopper-m-e|8.8|83.7|**97.1**|
> |halfcheetah-m-e|12.1|**87.4**|83.9|
> |walker2d-m-e|64.5|43.7|**108.6**|
> |Mean score|37.5|58.4|**70.6**|
>
>
> - We do not add more experiments on offline IL methods with different degrees of imperfection of rewards, because offline IL methods do not use rewards for policy learning.

---

> ### Author Response · Authors · 2022-11-12
> **Response to Reviewer KWzN W4**
>
> >**W4. Thus, I think there should be additional empirical studies on (1) perfect and incorrect rewards. (2) other types of imperfect rewards (e.g. adding i.i.d. gaussian noise for each reward).**
>
> We thank the reviewer for this comment. We have conducted additional experiments to address the concerns of the reviewer. The new results are also updated in Appendix E of our revised paper.
>
> - (1) We add experiments about perfect rewards and completely incorrect rewards. The results are listed below, see Appendix E for detailed setups. We flip all signs of perfect rewards to construct completely incorrect rewards (RGM 100% incorrect). RGM(T) means trained on perfect rewards. We can see RGM under perfect rewards slightly outperforms RGM with 50% incorrect rewards, which means RGM can largely remedy the negative effects caused by 50% incorrect rewards. Meanwhile, the totally incorrect rewards surely impact the performance, but its mean score is 41.9, which still outperforms other Offline RL and IL methods under 50% incorrect rewards with the largest mean value of 35.5 (see Table 1 in the main text).
>
> | Dataset        | RGM (T)   | RGM 50% incorrect (in our paper) | RGM 100% incorrect |
> | -------------- | --------- | -------------------------------- | ------------------ |
> | hopper-r       | **29.6**  | 21.2                             | 25.9               |
> | half-r         | 0.2       | 0.2                              | 0.2                |
> | walker2d-r     | 3.9       | **7.7**                          | -0.1               |
> | hopper-m       | **56.2**  | 55.5                             | 54.6               |
> | half-m         | 40.4      | **40.7**                         | 40.3               |
> | walker2d-m     | **73.3**  | 72.3                             | 38.4               |
> | hopper-m-r     | **60.3**  | 59.1                             | 44.5               |
> | half-m-r       | **37.9**  | 37.8                             | 29.8               |
> | walker2d-m-r   | 46.3      | **48.6**                         | 46.1               |
> | hopper-m-e     | **106.1** | 87.1                             | 66                 |
> | half-m-e       | **85.6**  | 81.5                             | 78.4               |
> | walker2d-m-e   | **109.2** | 108.8                            | 108.8              |
> | **Mean score** | **54.1**  | 52.0                             | 41.9               |
>
> - (2) We add i.i.d gaussian noise with different standard deviation $\sigma$ for original rewards. We set $\sigma=1$ as the biased reward setting and $\sigma=10$ as completely incorrect rewards. The results are listed as follows. We obtain a similar conclusion in (1) that RGM can robustly handle different degrees of imperfect rewards.
>
> | Dataset        | RGM (T)   | RGM with $\sigma=1$ | RGM with $\sigma=10$ |
> | -------------- | --------- | ------------------- | -------------------- |
> | hopper-r       | **29.6**  | 8.5                 | 9.8                  |
> | half-r         | 0.2       | **0.3**             | 0.2                  |
> | walker2d-r     | **3.9**   | 0.6                 | -0.1                 |
> | hopper-m       | **56.2**  | 52.0                | 47.9                 |
> | half-m         | 40.4      | **41.2**            | 38.4                 |
> | walker2d-m     | **73.3**  | 71.9                | 72.0                 |
> | hopper-m-r     | **60.3**  | 58.0                | 40.0                 |
> | half-m-r       | 37.9      | **38.3**            | 28.1                 |
> | walker2d-m-r   | **46.3**  | 42.5                | 43.8                 |
> | hopper-m-e     | **106.1** | 82.0                | 82.8                 |
> | half-m-e       | 85.6      | **88.7**            | 69.1                 |
> | walker2d-m-e   | **109.2** | 108.2               | 108.5                |
> | **Mean score** | **54.1**  | 49.4                | 45.0                 |
>
> - Additionally, we add sparse reward experiments on the extremely difficult Antmaze tasks. See response to the W6 of Reviewer 1KgB for detailed results.

---

> > ### Comment · Reviewer_KWzN · 2022-12-13
> > **Additional question and concern W4**
> >
> > I really thank the authors for further experiments.
> >
> > However, in many cases, incorrectness of reward function does not significantly affect to the performance.
> >
> > I think this is because the imperfect reward function only serves as an initialization of $\hat r$.

---

> > > ### Author Response · Authors · 2022-12-13
> > > **Addtional responses to W4**
> > >
> > > Please refer to our addtional responses to W1 and the Figure 5 in our paper.

---

> ### Author Response · Authors · 2022-11-12
> **Response to Reviewer KWzN W2~W3**
>
> > **W2. (Related to the previous question) Suppose that we can obtain optimal values for each optimization problem. In this ideal case, RGM obtains the optimal reward correction term, optimal stationary distribution, and optimal policy. Then, what is the main difference between RGM and existing offline IL algorithms in theory? I think RGM is equivalent to DemoDICE (if f-divergence is KL divergence) and SMODICE if the max / min can be computed exactly.**
>
> - The reviewer has raised a good point about the difference between RGM and existing DICE works. We provide some major differences between RGM and the two recent works (DemoDICE and SMODICE):
>   - In theory, DemoDICE considers the problem: $\max_{\pi} -D_{\mathrm{KL}}\left(d^\pi \| d^E\right)-\alpha D_{\mathrm{KL}}\left(d^\pi \| d^\mathcal{D}\right)$, minimizing the KL-divergence between $d^{\pi}$ and $d^E$ while keeping $d^\pi$ closed to $d^{\mathcal{D}}$. It leverages the offline dataset and expert dataset to formulate a "fake" reward as: $r(s,a)=-\log \left(\frac{d^\mathcal{D}(s,a)}{d^E(s,a)}\right)$ (Eq. (14) in DemoDICE paper). Similar to DemoDICE, SMODICE considers the problem: $\min_{\pi} D_{\mathrm{KL}}\left(d^\pi (s)\| d^E(s)\right)$ and instead optimizes the upper bound: $\mathbb{E}_{s \sim d^\pi}\left[\log \left(\frac{d^O(s)}{d^E(s)}\right)\right]+\mathrm{D}_f\left(d^\pi(s, a) \| d^O(s, a)\right)$, while formulating a "fake" reward as: $r(s)=-\log \left(\frac{d^{O}(s)}{d^E(s)}\right)$ (Eq.(12) in SMODICE paper). Both DemoDICE and SMODICE are essentially offline IL methods, that ignore or don't use the pre-collected rewards, even if the rewards can provide some useful information. Consequently, they both need a large amount of expert data to learn high-quality policies.
>   - In contrast, RGM fully utilizes the pre-collected imperfect rewards, which corrects wrong rewards against expert preference as well as retrieves useful information from given rewards. Essentially, RGM can be perceived as a hybrid algorithm with both offline IL and RL ingredients. We leverage offline IL ingredients for reward correction, rather than directly imitating expert policy as in typical offline IL methods. And with the corrected rewards, we perform offline RL for policy optimization on an arbitrary quality (potentially suboptimal) offline dataset. This design allows using minimal expert data and imperfect rewards for effective policy learning.
>   - Furthermore, DemoDICE and SMODICE optimize a single-level IL objective, but RGM optimizes a bi-level optimization problem where $d^{\pi^{*}_{\hat{r}}}$ in the upper level divergence minimization objective (Eq.(6)) comes from the optimal policy of lower level RL problem (Eq.(7)).
>   - We also include some illustrated results about the differences between RGM and DemoDICE/SMODICE in Appendix C.2 of our revised paper. The results show that without the ability to leverage useful information in the provided imperfect rewards, DemoDICE and SMODICE are unable to provide correct learning signals in no-expert-covered regions, whereas RGM can provide correct learning signals even in regions not covered by the expert data.
>
> >**W3. ...therefore, in my opinion, learnable reward correction term also can provide correct signals when dE>0.**
>
> - In continuous MDPs, both our implementation and SMODICE/DemoDICE implementations add gradient penalty into discriminator loss to avoid output $0$ and $\infty$. So, the rewards in SMODICE/DemoDICE and $w:=\frac{d^E}{d^ D}$ in RGM are valid numbers. However, due to function approximation errors, $\log\frac{d^E}{d^D}$ and $w$ will be too small and noisy to provide meaningful information in no-expert-covered regions. However, the upper level of RGM solves a distribution matching problem, making the learned reward not only focuses on current state-action pairs, but also backpropagates some useful long-term information through $\hat{r}(s,a)+\gamma \mathcal{T}V^*(s,a)-V^*(s)$ (analogous to some form of "bellman error") in Eq (17). In contrast, no similar mechanism is designed in SMODICE and DemoDICE and hence their rewards are meaningless in non-expert regions.
>
> - Specifically, we add a more illustrative example in Figure 7 in Appendix C.2 in the revised paper. Take the grid world in Figure 6 as an example, we set $\sum_{(\overline s,\overline a)\in\mathcal{D}^E}\mathbb{I}(\overline s=s, \overline a=a)=1$ when $d^E(s,a)=0$ in practice to avoid invalid number like $\log(0)$. Figure 7 shows that the reward signal of SMODICE and DEMODICE is not distinguishable for the  actions in non-expert regions, while the learned rewards of RGM can encode long horizon information, correct wrong rewards against expert preference, and extract useful information from existing rewards. Please check the detailed discussion in Appendix C.2 in our revised paper.

---

> > ### Comment · Reviewer_KWzN · 2022-12-13
> > **Additional question and concern W2-3**
> >
> > In general, KL-divergence matching can be expressed as an RL problem where rewards are given by log distribution ratio:
> >
> > $-D\_\text{KL}(d^\pi\\|d^E)=\mathbb{E}\_{d^\pi}\Big[\log\frac{d^E(s,a)}{d^\pi(s,a)}\Big]$
> >
> > Based on the formulation, previous online IL algorithms solve the problem by repeating the following two steps:
> >
> > 1. For given $r(s,a)$, solve $\max\_\pi \mathbb{E}\_{d^\pi}[r(s,a)]$ (+ regularization if needed)
> > 2. For given $\pi$, compute $r(s,a)=\frac{d^E(s,a)}{d^\pi(s,a)}$.
> >
> > Similarly, when $f$-divergence is a KL divergence, RGM solves the problem by repeating the following two steps:
> >
> > 1. (Eq. (7)) For given $\hat r$, obtain $\pi^*\_{\hat r}$ by solving $\max_\pi\mathbb{E}\_{d^\pi}[\hat r(s,a)]$ (+ small regularization term)
> > 2. (Eq. (6)) For given $\pi$ (obtained in step 1.), compute $\hat r(s,a)$ as $\hat r(s,a)=\frac{d^E(s,a)}{d^\pi(s,a)}$
> >
> > And to improve the stability, the previous offline IL algorithms construct single convex optimization using $\log\frac{d^E(s,a)}{d^D(s,a)}$.
> > In this perspective, the main difference between RGM and previous offline IL algorithms is:
> >
> > - RGM computes $\log\frac{d^E(s,a)}{d^\pi(s,a)}$ directly using Eq. (6) and optimize policy using Eq. (7).
> > - DICE-based algorithms compute $\log\frac{d^E(s,a)}{d^\pi(s,a)}=\log\frac{d^E(s,a)}{d^D(s,a)}-\log\frac{d^\pi(s,a)} {d^D(s,a)}$ and optimize policy simultaneously using single convex optimization (using pretrained $\log\frac{d^E(s,a)}{d^D(s,a)}$).
> >
> > Consequently, I cannot agree that the reward learning and policy optimization proposed in RGM are better than previous offline IL algorithms.

---

> > > ### Author Response · Authors · 2022-12-13
> > > **Additional responses to W2-3**
> > >
> > > **The following statements mentioned by the reviewer are incorrect.**
> > >
> > > >(Eq. (6)) For given (obtained in step 1.), compute $\hat{r}(s,a)$ as $\hat{r}(s,a)=\frac{d^E(s,a)}{d^\pi(s,a)}$.
> > >
> > > >RGM computes $\log\frac{d^E(s,a)}{d^\pi(s,a)}$ directly using Eq. (6) and optimize policy using Eq. (7)
> > >
> > >
> > > **It seems that there might be some misunderstanding by the reviewer on the methodology of our paper.** The optimal reward of RGM is not $\frac{d^E(s,a)}{d^\pi(s,a)}$, but comes from the optimal solution of the upper level problem Eq.(6) : $\Delta r^* =\arg \min_{\Delta r} D_f(d^{\pi^*_{\hat{r}}}||d^E)$, where $d^{\pi^*_{\hat{r}}}$ is the optimal solution of Eq. (7). It's obvious that these two terms are different. So, RGM is very different from the previous online IL and DICE-based methods mentioned by the reviewer.
> > >
> > > The experiments on the grid world environment in Figure 3(b), (c) also illustrate that RGM can retrieve useful information from the given imperfect rewards $\tilde{r}$ (e.g., avoid the fire region according to the information only provided in the imperfect rewards $\tilde{r}$), which cannot be achieved by offline IL methods mentioned by the reviewer that solely imitate the expert data.

---

> ### Author Response · Authors · 2022-11-12
> **Response to Reviewer KWzN W1**
>
> We sincerely appreciate the reviewer for the thoughtful comments. Below are detailed responses to each comment:
>
> > **W1. RGM does not consider the degree of imperfection of rewards. Theoretically, it seems that the degree has no effect on the performance of RGM - it is weird to me. Intuitively, if rewards are perfect, RGM should perform better than offline IL algorithms and comparable to offline RL algorithms, and if rewards are too imperfect, RGM should perform poorly. Consequently, it is questionable whether the imperfection of rewards affects RGM performance.**
>
> - In fact, RGM implicitly considers the degree of imperfection of rewards. Note that the inputs of RGM algorithm consist of a pre-collected dataset $\mathcal{D}^O$ with potentially imperfect rewards $\tilde{r}$ and a small expert dataset $\mathcal{D}^E$. With the help of $\mathcal{D}^E$, RGM can effectively correct wrong or overly imperfect rewards against expert preferences. Here, the discrepancy of patterns in $\mathcal{D}^E$ and imperfect rewards $\tilde{r}$ reflects the degree of imperfection of rewards. Essentially, RGM aims to minimize the $f$-divergence between $d^{\pi^*_{\hat{r}}}$ and $d^E$ in Eq.(6) through the learnable $\Delta r$ in $\hat{r}$ ($\hat{r}=\tilde{r}+\Delta r$). So even if given an overly imperfect rewards $\tilde{r}$, the learnable $\Delta r$ adjusts $\hat{r}$ to approximate the perfect reward $r$ and make the distribution $d^{\pi^{*}\_{\hat{r}}}$ approximate to $d^E$. As a result, RGM can also achieve good performance even if $\tilde{r}$ is not good. On the other hand, RGM is designed to handle diverse reward settings, so we choose to not add additional assumptions or restrictions on the degree of imperfection of the pre-collected rewards $\tilde{r}$.
> - Essentially, RGM can be perceived as a hybrid algorithm with both offline IL and RL ingredients. We leverage offline IL ingredients to correct wrong rewards as well as retrieve useful information from given rewards, rather than directly imitating expert policy as in typical offline IL methods. And with the corrected rewards, we perform offline RL for policy optimization on an arbitrary quality (potentially suboptimal) offline dataset. This design allows using minimal expert data and imperfect rewards for effective policy learning.

---

> > ### Comment · Reviewer_KWzN · 2022-12-13
> > **Additional question and concerns W1**
> >
> > In Eq. (6-7), learning $\Delta r$ is equivalent to learning $\hat r$ because $\tilde r$ is a given function. An optimal solution of $\hat r(s,a)=r(s,a)$, where $r(s,a)$ is a true reward function.
> > It means that the role of $\tilde r$ is an initialization of $\hat r$.
> > In practice, a good $\tilde r$ can accelerate $\hat r$ learning, but in theory, $\tilde r$ does not affect the optimal solution of $\hat r$. Thus, as authors mentioned, RGM can achieve good performance even if $\tilde r$ is not good.
> >
> > In addition, not only $r(s,a)$, but also $c\cdot r(s,a)$ where $c\in\mathbb{R}\_{>0}$ can be a solution of $\hat r(s,a)$. So RGM also an ill-posed problem. It is a very important issue in inverse RL community, but there is no discussion on this issue.

---

> > > ### Author Response · Authors · 2022-12-13
> > > **Addtional responses to W1**
> > >
> > > Actually, in RGM, $\tilde{r}$ does affect the learned optimal solution of $\hat{r}$. In Section 5.3, we have conducted detailed investigations on the impacts of $\tilde{r}$ on the learned rewards $\hat{r}$ in both a grid world experiment and the Hopper-m-r task. All these experiments suggest that different input imperfect rewards $\tilde{r}$ could result in different patterns of $\hat{r}$. In short, the learned rewards of RGM enjoy three desirable properties: 1) encode long horizon information; 2) correct wrong rewards against expert preference; and 3) retrieve useful information from existing rewards. Such properties are impossible to be obtained in previous IL methods that solely imitate expert behavior. Please refer to Section 5.3 in our paper for details.
> > >
> > > >In addition, not only $r(s,a)$, but also $c\cdot r(s,a)$ where $c\in\mathbb{R}_{>0}$ can be a solution of $\hat{r}(s,a)$. So RGM also an ill-posed problem. It is a very important issue in inverse RL community, but there is no discussion on this issue.
> > >
> > > In fact, multiplying the reward function by a constant factor $c\in\mathbb{R}_{>0}$ also impacts policy learning when using function approximators, as discussed in the "Perfect Rewards" part of Related Work in our submitted paper. Briefly, a recent NeurIPS paper [1] studies this phenomenon and finds that the linear transformation of rewards is equivalent to engineering the initialization of Q-function estimation. A positive reward shifting leads to conservative exploitation, while a negative reward shifting leads to curiosity-driven exploration [1]. Hence an improperly scaled reward function may act similarly to an imperfect reward that leads to over-conservative or over-aggressive sub-optimal policies. Only a properly scaled reward function can guarantee the best policy learning performance. As RGM adopts a bi-level optimization framework, with the lower level problem finds the optimal policy of an RL problem, hence RGM actually automatically learns to properly scale the reward that can best facilitate policy learning rather than admitting multiple optimal solutions of $\hat{r}$. As illustrated in **Fig. 5**, the reward learned by RGM actually stably converges to a unique scale of the reward.
> > >
> > > [1] Hao Sun, Lei Han, Rui Yang, Xiaoteng Ma, Jian Guo, and Bolei Zhou. Optimistic Curiosity Exploration and Conservative Exploitation with Linear Reward Shaping . In Advances in Neural Information Processing Systems, 2022.

---

> ### Author Response · Authors · 2022-11-22
> **Follow up**
>
> Dear reviewer KWzN,
>
> >However, I have some questions and I believe they are important in evaluating this paper. Thus, I vote to weakly reject until these questions are resolved.
>
> Thank you for the detailed comments and suggestions to improve our paper! Regarding your concerns, we have added more experiments and discussions in our latest revision. Hope our clarifications and responses have addressed your concerns. We would really appreciate it if you could tell us if your concerns are resolved. If you have further concerns or questions, please tell us; they will greatly help to improve the quality of our paper and we will be more than happy to engage in further discussions and address them.
>
> Thanks！

---

> ### Author Response · Authors · 2022-12-11
> **Sorry to bother you!**
>
> Dear reviewer KWzN,
>
> >However, I have some questions and I believe they are important in evaluating this paper. Thus, I vote to weakly reject until these questions are resolved.
>
> As the discussion period is coming to a close, we wanted to check back to see whether you have any remaining questions. **We would be happy to clarify further, and grateful for any other feedback you may provide.** We really appreciate your time engaged in the review and rebuttal phase.
>
> Thank you very much and look forward to your replies!
>
> Best regards,
>
> Authors of Paper 448

---

> > ### Comment · Reviewer_KWzN · 2022-12-13
> > **Sorry for the late feedback**
> >
> > I am really sorry for the late feedback and thank you for the detailed response.

---

> > > ### Author Response · Authors · 2022-12-13
> > > **Thanks for the replies**
> > >
> > > Dear reviewer KWzN,
> > >
> > > Thanks for your time engaged in the discussion phase. But unfortunately, there are only a few hours left for the rebuttal phase 2. As a result, we have no time to carry out the further detailed discussion.
> > >
> > > We respectfully disagree with some points you mentioned. We have tried our best to address your concerns in a concise way. Hope our additional responses address your concerns.
> > >
> > >
> > > Best,
> > >
> > > Authors of Paper 448

---

### Official Review · Reviewer_1KgB · 2022-10-27

**Confidence:** 5
**Correctness:** 3
**Technical Novelty And Significance:** 2
**Empirical Novelty And Significance:** 2
**Recommendation:** 5

**Clarity, Quality, Novelty And Reproducibility:**

**Quality**

I believe there are some technical flaws in the paper, which require the authors to clarify.


**Clarity**

The paper is well written and is easy to follow the derivation.

**Originality**

As far as I know, there are no previous work utilizes density ratio estimation for imperfect reward learning and policy improvement.

**Strength And Weaknesses:**

**Strength**
- The authors conducted detailed empirical experiments to demonstrate the advantage of their approach under the offline settings.
- The derivations in the paper are quite comprehensive, which are good for authors to follow in detail, and check out how the results are derived.


**Weakness**
1. I believe there are some fundamental technical faults in the basic formulation of the paper, regardless of the superior empirical performance.  Here are some technical flaws I am concerned about:
      - In Equation (5) & (6), you define the gap between the current reward function and the optimal reward function as $$D_{f}( d^{\pi^{*}}_{\hat{r}} | d^{E}) $$

     According to your definition, $ d^{\pi^{*}}_{\hat{r}}$ is the  state-action visitation distribution defined in Equation (4). In Equation (4), I believe the discounted visitation distribution is defined together with $\gamma$ and initial distribution. As I can infer from your following derivation, the stationary distribution of the expert data $d^{E}$ you used is actually the sampling distribution, which is not related to the discounted factor $\gamma$ and initial distribution. If so, how could you actually minimize the gap? They are two different kinds of stationary distribution. Let me illustrate a more intuitive case, suppose $\gamma$ is very small, close to zero (e.g., $\gamma=0.01$), your expert distribution is define on the whole trajectory (unrelated to $\gamma$), while the stationary distribution of the policy obtained by the reward function $\hat{r}$ is actually close to the initial state distribution and the policy distribution, then your reward gap definition is totally problematic.

  Similar thing happens when you try to estimate Eq (16). The density ratio you estimated is the undiscounted ($\gamma=1$ or sampling) distribution, while in your whole derivations $d$ is the discounted distribution. I feel it is not correct to match the discounted stationary distribution to the undiscounted stationary (sampling) distribution, unless you proof that it is okay to do that, and you can convince me the previous example I illustrated is wrong.

3. For estimating Eq(13) or Eq(43), I am wondering how would actually deal with double sampling? your choice of conjugate function does now allow you to obtain unbiased estimation since there is an expectation in $TV^{*}(s)$? Do you assume the transitions and policy is deterministic? If so, I do not see any related assumptions in your paper.

4. The whole algorithm is too complex, I feel it may require lots of efforts and hyperparameter tunning to make it work.

5. Why you want to conduct the exepriments in the offline setting? I believe the imperfect problem issue exists in all settings of RL.

6. And also, how do you define **imperfect** and **perfect** reward in the mujoco environments? the mujoco reward functions you used are human designed, why would you think they are perfect. One case I can think about is that in the sparse reward settings (or multi-goal RL), the perfect reward is that you would achieve higher success rate when testing. So I don't think mujoco is the perfect environments to demonstrate your claim, since there is no definition of perfect reward here (correct me if I am wrong, this is my personal thoughts).


**Summary Of The Paper:**

This paper propose an approach to address the problem of imperfect reward under offline settings.

The main idea is to formulate the problem as a bi-level optimization problem, where the upper problem is to minimize the gap between the stationary distribution of the optimal policy defined by the imperfect reward, and the distribution of the expert data. The lower optimization problem is to use the optimized reward function to obtain a new policy under the offline RL settings. To track the bi-level optimization problem, the authors leverage the fenchel duality to obtain a tractable optimization solution.

Empirically the authors demonstrate that their approach can obtain better performance when the reward function is imperfect.

**Summary Of The Review:**

overall I think the paper is not ready in terms of the following two main reasons:

- There are fundamental issues of their reward gap and followed up derivations.
- The experiments they conducted I think is not the right settings to demonstrate the advantage of their approach. In most of the scenarios that perfect reward functions are hard to obtain, such as sparse reward settings, they did nothing on that.

As a result, I recommend a major revision of the paper.

For authors' rebuttal message: I know I am quite strict about your definition and derivation details, but please correct me if there is anything wrong of my points. I am very happy to modify scores if I miss anything important.

----

Typos:
- Line under Eq(11), it should be $TV(s)$, not $TV(s,a)$

---

> ### Author Response · Authors · 2022-11-12
> **Reference**
>
> [1] Mnih, Volodymyr, et al. "Asynchronous methods for deep reinforcement learning." *International conference on machine learning*. PMLR, 2016.
>
> [2] Wang, Ziyu, et al. "Sample efficient actor-critic with experience replay." *In Proceedings of the 5th International Conference on Learning Representations*. 2017.
>
> [3] Wu, Yuhuai, et al. "Scalable trust-region method for deep reinforcement learning using kronecker-factored approximation." *Advances in neural information processing systems* 30 (2017).
>
> [4] Lillicrap, Timothy P., et al. "Continuous control with deep reinforcement learning." *In Proceedings of the 4th International Conference on Learning Representations*. 2016.
>
> [5] Schulman, John, et al. "Trust region policy optimization." *International conference on machine learning*. PMLR, 2015.
>
> [6] Schulman, John, et al. "Proximal policy optimization algorithms." *arXiv preprint arXiv:1707.06347* (2017).
>
> [7] Fujimoto, Scott, Herke Hoof, and David Meger. "Addressing function approximation error in actor-critic methods." *International conference on machine learning*. PMLR, 2018.
>
> [8] Haarnoja, Tuomas, et al. "Soft actor-critic: Off-policy maximum entropy deep reinforcement learning with a stochastic actor." *International conference on machine learning*. PMLR, 2018.
>
> [9] Ho, Jonathan, and Stefano Ermon. "Generative adversarial imitation learning." *Advances in neural information processing systems* 29 (2016).
>
> [10] Nachum, Ofir, et al. "Dualdice: Behavior-agnostic estimation of discounted stationary distribution corrections." *Advances in Neural Information Processing Systems* 32 (2019).
>
> [11] Nachum, Ofir, et al. "Algaedice: Policy gradient from arbitrary experience." *arXiv preprint arXiv:1912.02074* (2019).
>
> [12] Kostrikov, Ilya, Ofir Nachum, and Jonathan Tompson. "Imitation learning via off-policy distribution matching." *In Proceedings of the 8th International Conference on Learning Representations*. 2020.
>
> [13] Lee, Jongmin, et al. "Optidice: Offline policy optimization via stationary distribution correction estimation." *International Conference on Machine Learning*. PMLR, 2021.
>
> [14] Kim, Geon-Hyeong, et al. "DemoDICE: Offline imitation learning with supplementary imperfect demonstrations." *International Conference on Learning Representations*. 2021.
>
> [15] Ma, Yecheng Jason, et al. "SMODICE: Versatile Offline Imitation Learning via State Occupancy Matching."  *International Conference on Machine Learning*. PMLR, 2022.
>
> [16] Thomas, Philip. "Bias in natural actor-critic algorithms." *International conference on machine learning*. PMLR, 2014.
>
> [17] Nota, Chris, and Philip S. Thomas. "Is the Policy Gradient a Gradient?." *Proceedings of the 19th International Conference on Autonomous Agents and MultiAgent Systems*. 2020.
>
> [18] Baird, Leemon. "Residual algorithms: Reinforcement learning with function approximation." *Machine Learning Proceedings 1995*. Morgan Kaufmann, 1995. 30-37.
>
> [19] Antos, András, Csaba Szepesvári, and Rémi Munos. "Learning near-optimal policies with Bellman-residual minimization based fitted policy iteration and a single sample path." *Machine Learning* 71.1 (2008): 89-129.
>
> [20] Farahmand, Amir-massoud, et al. "Regularized policy iteration with nonparametric function spaces." *The Journal of Machine Learning Research* 17.1 (2016): 4809-4874.
>
> [21] Feng, Yihao, Lihong Li, and Qiang Liu. "A kernel loss for solving the bellman equation." *Advances in Neural Information Processing Systems* 32 (2019).
>
> [22] Nachum, Ofir, and Bo Dai. "Reinforcement learning via fenchel-rockafellar duality." *arXiv preprint arXiv:2001.01866* (2020).

---

> ### Author Response · Authors · 2022-11-12
> **Response to Reviewer 1KgB W5~W6**
>
> > **W5. Why you want to conduct the exepriments in the offline setting? I believe the imperfect problem issue exists in all settings of RL.**
>
> - Actually, our method can also be applied to the online setting. By simply setting $\alpha=0$ in Eq.(6-7), we have the bi-level objective of RGM (online version):
>
>
> $$\quad\quad\quad\Delta r^*=\arg\min_{\Delta r} D_f\left(d^{\pi^*_{\hat{r}}} \\|d^E\right)  \quad \text{s.t.} \pi^{*}\_{\hat{r}} = \underset{\pi}{\arg \max} \mathbb{E}\_{(s, a) \sim d^{\pi\_{\hat{r}}}}[\hat{r}(s,a)]
> $$
>
> - Since we could get online samples from $d^{\pi^*_{\hat{r}}}$ in the online setting, so we don't have to eliminate $d^{\pi^*_{\hat{r}}}$. One can use the existing popular online RL algorithms to solve the lower-level problem while leveraging the online samples from  $d^{\pi^*_{\hat{r}}}$ to solve the upper-level problem. The core idea of the reward correction has not been changed in the online setting, which illustrates that to some extent, our RGM is a unified policy optimization method for imperfect rewards. The reason why we care more about the offline setting in this paper is that:
>
>   - The offline setting is more realistic in many real-world applications, where we cannot get the online samples during policy training due to system safety and cost issues. As both offline learning and dealing with imperfect rewards are two major challenges in real-world RL practices, we were motivated to address both two challenges with a single new framework.
>   - Reward correction or reward shaping is far more difficult in offline setting than that in online setting. We would like to be the pioneer to propose a unified offline policy optimization approach for diverse settings of imperfect rewards.
>
> >**W6. And also, how do you define imperfect and perfect reward in the mujoco environments?...**
>
> - It is worth noting that there should be no reward gap if we train and test the policy using the same reward function. Given a specific task, there may be one perfect reward that can describe this task. Meanwhile, given a specific reward function, there is also a corresponding task described by this reward. So, although the mujoco rewards are human-designed, the rewards can be perceived as perfect because we test our policy using the same reward function.
> - As a matter of fact, the 8×8 grid world navigation experiment in Section 5.3 and Appendix C.2 is a sparse reward task, because the agent gets no reward unless it gets to the destination or fires. In addition, we add additional experiments on extremely difficult Antmaze sparse reward tasks, where an ant must learn to walk and simultaneously navigate to the goal in a maze guided by a sparse reward located at the goal. Note that in typical offline RL papers, like IQL and CQL, they turn the original sparse rewards into dense rewards by applying the reward subtraction trick (minus 1 on every reward, so the reward becomes negative except at the goal). These offline RL methods can obtain non-zero successful rates with the substracted dense rewards but fail miserably when trained on the original sparse reward. In this experiment, we train IQL using dense rewards to collect expert data to serve as $\mathcal{D}_E$. The successful rates of RGM and other offline RL methods are listed below, see detailed experimental setups in Appendix E.6 of our revised paper. The results show that RGM is the only approach that can solve such extremely difficult sparse reward tasks.
>
> | Dataset     | TD3+BC | CQL  | IQL  | RGM(ours) |
> | ----------- | ------ | ---- | ---- | --------- |
> | Antmaze-m-p | 0      | 0    | 0    | 13.7%         |
> | Antmaze-m-d | 0      | 0    | 0    | 3.3%         |
>
>
> > Typos: Line under Eq(11), it should be $\mathcal{T}V(s)$, not $\mathcal{T}V(s,a)$
>
> - We thank the reviewer for pointing out the typo. It should be $\mathcal{T}V(s,a)=\sum_{s'}T(s'|s,a)V(s')$. We have revised it.

---

> ### Author Response · Authors · 2022-11-12
> **Response to Reviewer 1KgB W2~W4**
>
> > **W2. Similar thing happens when you try to estimate Eq (16). The density ratio you estimated is the undiscounted ($\gamma$=1 or sampling) distribution, while in your whole derivations $d$ is the discounted distribution. I feel it is not correct to match the discounted stationary distribution to the undiscounted stationary (sampling) distribution, unless you proof that it is okay to do that, and you can convince me the previous example I illustrated is wrong.**
>
> - Please refer to our response of W1, the distribution ratio $w(s,a)$ is actually a ratio of two discounted visitation distributions. In practical implementation, we empirically estimate $w(s,a)$​ for the tubular case while adopting sampling distribution in Eq.(16) for the continuous state-action settings.
>
> > **W3. For estimating Eq(13) or Eq(43), I am wondering how would actually deal with double sampling? your choice of conjugate function does now allow you to obtain unbiased estimation since there is an expectation in $TV^{*}(s)$? Do you assume the transitions and policy is deterministic? If so, I do not see any related assumptions in your paper.**
>
> - We thank the reviewer for this thoughtful comment. The appearance of $TV^*$ inside the conjugate function in Eq.(13), (17) and (43) presents a double sampling challenge and a biased estimation, which is also described by some RL works [18]. Some techniques are proposed to handle or mitigate this problem [13,19-21], including the dual embedding technique adopted by AlgaeDICE [11]. However, the later ValueDICE paper [12] claims that handling double sampling entangled with the Fenchel conjugate might be unnecessary (see Section 5.1 of [12]) and instead uses a biased estimation based on the single sample, which is enough to achieve a good performance. Many recent related works [13,15,22] also do not find this biased estimate to impact empirical performance and keep it for simplicity. Our empirical results also demonstrate that it might be okay to implement with the simple sample-based estimation.
>
> > **W4. The whole algorithm is too complex, I feel it may require lots of efforts and hyperparameter tunning to make it work.**
>
> - As illustrated in Figure 1, the whole algorithm is separated into 3 **decoupled** learning procedures. As a result, it suffers less from the bootstrapping error accumulation issue in typical actor-critic-based offline RL methods.
> - In practice, the only hyperparameter unique in RGM is the conservative coefficient $\alpha$ in Eq.(7), which is set to 0.5 for almost all 12 mujoco tasks, which means our method is also robust to hyperparameter tuning.
> - Moreover, offline policy optimization under imperfect rewards is itself a far more complex and challenging task compared with standard offline RL tasks. Under imperfect rewards, we need to optimize offline RL policies while combating the negative impacts of imperfect rewards. Hence it is reasonable to expect some additional designs to address this challenge.

---

> ### Author Response · Authors · 2022-11-12
> **Response to Reviewer 1KgB Weakness 1 (Part 1/3)**
>
> We sincerely thank the reviewer for the detailed comments.  Below are detailed responses to each comment:
>
> > **W1. I believe there are some fundamental technical faults in the basic formulation of the paper, regardless of the superior empirical performance. Here are some technical flaws I am concerned about...**
>
> - We respectfully disagree with the reviewer's opinion on the "inconsistent" reward gap definition. For clarity, we follow the reviewer's expression and refer the **discounted visitation distribution** as the distribution defined in Eq.(3) related to $\gamma$ and the initial distribution; and **sampling distribution** as the distribution defined without the discounted factors.
>
>   - In the definition of reward gap (Eq.(5)), the optimal state-action visitation distribution $d^*$ is actually the **discounted visitation distribution** of the optimal policy induced from the perfect rewards $r$. We use $d^E$ induced from the expert policy $\pi^E$ to approximate $d^*$, while $d^E$ is also the **discounted visitation distribution**. In fact, the two distributions in the reward gap definition are consistent and they both contain the discounted factor. In the theoretical derivation of this paper (Section 4.1 and 4.2), we have always been regarding $d^{\pi^*_{\hat{r}}}$, $d^*$,$d^E$, $d$ and $d^\mathcal{D}$ as the **discounted visitation distribution**. In theory, $(s,a)\sim d^E$ and $(s,a)\sim d^D$ in Eq.(16-18) mean that the state-action pairs are sampled from the **discounted visitation distribution** $d^E$ and $d^D$.
>
>   - Given the fact $d^E$ is actually a discounted visitation distribution, It might be easy to find that the example the reviewer illustrates is inappropriate. When $\gamma$ is very small, $d^E$ is also close to the initial state distribution.
>
>   - We conjecture that some expressions in our paper may lead to a misunderstanding of the two different distributions for the reviewer. For example, the expression "$d^E$ in expert demonstrations $\mathcal{D}^E$ " below Eq.(5) actually means that the dataset $\mathcal{D}^E$ is generated by an unknown expert policy $\pi^E$, while $d^E$ is the discounted visitation distribution induced from $\pi^E$, instead of the sampling distribution. We have modified such confusing expressions in our paper. We thank the reviewer for the comment on the two different distributions, but there exist no theoretical flaws about the two distributions in our method.

---

> > ### Author Response · Authors · 2022-11-12
> > **Response to Reviewer 1KgB Weakness 1 (Part 2/3)**
> >
> > - For practical implementation of Eq.(16-18) in our algorithm, we choose to sample state-action pairs from the **sampling distribution** $E(s,a)$ and $D(s,a)$ instead of the **discounted visitation distribution** $d^E(s,a)$ and $d^D(s,a)$ for simplicity. If we denote the term in the expectation under a discounted visitation distribution $d^{\pi}$ as $F(s,a)$, then $\mathbb{E}\_{s,a\sim d^{\pi}}[F(s,a)]$ can be equivalently written as  $(1-\gamma) \mathbb{E}\_{s\_0\sim \mu\_0, a_t\sim \pi, s\_{t+1}\sim T} \left[\sum_{t=0}^{\infty} \gamma^tF(s_t,a_t)\right]$. Since we work on gradient-based optimization, the multiplier $1-\gamma$ can be typically incorporated into the learning rate. For practical implementation, we favor the sampling distribution, which is equivalent to drop $\gamma^{t}$ (or set $\gamma= 1$) from $\sum_{t=0}^{\infty} \gamma^tF(s_t,a_t)$ while remaining $\gamma$ inside $F(s\_t,a\_t)$. We highlight that it is prevalent in copious previous works [1-15] to drop the discount factor from the discounted visitation distribution in exchange for simple practical implementation, even in many well-known algorithms in RL community, exactly as what we did in the practical implementation of RGM. We have surveyed some well-known works and implementations on this issue:
> >
> >     - As shown by [16-17], A2C/A3C, ACER, ACKTR, DDPG, TRPO, PPO, TD3 and SAC ignore the discount factor from the discounted visitation distribution, both in their pseudocode and implementations (stable-baselines library and the official codes from the authors). These algorithms are developed based on the policy gradient theorem:
> >
> >     $$
> >       \nabla_\theta J\left(\pi_\theta\right) =\mathbb{E}\_{s\_0 \sim \mu\_0, a_t\sim \pi, s\_{t+1}\sim T}\left[\sum\_{t=0}^{\infty} \gamma^t \nabla\_\theta \log \pi\_\theta\left(a_t | s\_t\right) Q^\pi\left(s_t, a_t\right)\right] = \frac{1}{1-\gamma}\mathbb{E}_{(s,a) \sim d^{\pi}}\left[\nabla_\theta \log \pi_\theta\left(a | s\right) Q^\pi\left(s, a\right)\right]
> >     $$
> >
> >     - However, all of the aforementioned algorithms instead use the following expression as the policy gradient:
> >
> >     $$\qquad\qquad\mathbb{E}\_{s_0 \sim \mu_0, a_t\sim \pi, s_{t+1}\sim T}\left[\sum_{t=0}^{\infty} \nabla_\theta \log \pi_\theta\left(a_t | s_t\right) Q^\pi\left(s_t, a_t\right)\right]$$
> >
> >     - They actually adopt the sampling distribution instead of discounted visitation distribution, but remain the discounted factor in the state-action value function $Q^{\pi}$. In the practical implementation of Eq.(16-18), we also adopt the sampling distribution.
> >
> >    - In the fields of imitation learning and inverse reinforcement learning, plenty of works [9-15] have adopted the sampling distribution in place of the discounted visitation distribution for practical implementation, but remain $\gamma$ in $Q^{\pi}$ or $V^{\pi}$, although their methods are based on the discounted visitation distribution matching during theoretical derivation.

---

> > > ### Author Response · Authors · 2022-11-12
> > > **Response to Reviewer 1KgB Weakness 1 (Part 3/3)**
> > >
> > > - To further address the concerns of the reviewer, we have also implemented the discounted visitation distribution sampling in RGM (see Appendix E.2 in our revised paper). This is done by augmenting the D4RL datasets that add the timestep of each $(s,a)$ pair in an episode. When performing sampling in Eq.(16-18) and calculating the gradient, we sample $(s,a,t)$ in the D4RL datasets and then multiply the $F(s,a)$ by $\gamma^t$, and keep the $\gamma$ in $\gamma^t$ the same value as $\gamma$ inside $F(s,a)$. Empirically, we found that the performance of discounted visitation distribution version is not better than the sampling distribution version of RGM. As we can see from the following table, RGM (sampling distribution) surpasses RGM (discounted visitation distribution) in most cases, while the latter wins by a slight margin in only one dataset (please check Appendix E.2 in our revised paper for more detailed experiment results). In the following, we present some underlying reasons why RGM (sampling distribution sampling) could perform better in practice. In sampling distribution, every state-action pair along the episode is treated equally, while in discounted visitation distribution, we care more about the early state-action pairs than the state-action sampled far into the episode. Although it is unbiased to sample from the discounted visitation distribution, the agent tends to "over-optimize" on early states, while paying too little attention to later states, and it becomes even worse when certain later states are crucial for policy learning. What's more, when evaluating the policy, we usually look at the undiscounted returns, where the impact of the actions on the later states has not been discounted. So in practice, the sampling distribution mitigates the overlook of the later states and enjoys a better performance during evaluation.
> > >
> > >     | D4RL Dataset | RGM (discounted visitation distribution) (B) | RGM (sampling distribution) (B) |
> > >     | ------------ | :------------------------------------------: | ---------------------------------------- |
> > >     | hopper-r     |                     19.8±10.2                | **21.2**±0.4                             |
> > >     | halfcheetah-r|                     **0.2**±0.0              | **0.2**±0.0                              |
> > >     | walker2d-r   |                     1.2±1.7                  | **7.7**±3.3                                  |
> > >     | hopper-m     |                     51.1±4.9                 | **55.5**±1.0                                 |
> > >     | halfcheetah-m|                     40.3±1.6                 | **40.7**±1.4                                 |
> > >     | walker2d-m   |                     62.2±22.5                | **72.3**±10.7                                |
> > >     | hopper-m-r   |                     43.3±11.6                | **59.1**±15.3                                |
> > >     | halfcheetah-m-r |                  34.5±4.5                 | **37.8**±2.6                                 |
> > >     | walker2d-m-r |                     34.3±11.0                | **48.6**±3.6                                 |
> > >     | hopper-m-e   |                     65.3±19.5                | **87.1**±10.7                                |
> > >     | halfcheetah-m-e |                  **87.3**±7.8             | 81.5±0.9                                     |
> > >     | walker2d-m-e |                    108.4±0.6                 | **108.8**±0.4                                |
> > >     | Mean score   |                     45.7±7.2                 | **52.0**±4.2                                 |

---

> ### Author Response · Authors · 2022-11-22
> **Follow up**
>
> Dear reviewer 1KgB,
> > I know I am quite strict about your definition and derivation details, but please correct me if there is anything wrong of my points. I am very happy to modify scores if I miss anything important.
>
> Thank you for the detailed comments and suggestions to improve our paper! Regarding your concerns, we have added the discussions and more experiments in our latest revision. Hope our clarifications and responses have addressed your concern. We would really appreciate it if you could tell us whether your concerns are resolved. If you have further concerns or questions, please tell us; they will greatly help to improve the quality of our paper and we will be more than happy to engage in further discussions and address them.
>
> Thanks！

---

> ### Comment · Reviewer_1KgB · 2022-11-22
> **Thank the authors for the Detailed Response.**
>
> I would like to thank the authors for the detailed response. Here are some further comments:
>
> 1.
>  > We respectfully disagree with the reviewer's opinion on the "inconsistent" reward gap definition.
>
> My opinion is based on your original paper draft. In the submitted version, you haven't distinguished the difference between discounted stationary (visitation) distribution. I believe I had checked the definition and terminology in your original draft before I submitted the review. Actually, without the clarification of the difference, if a beginner who is new to this field and he takes a quick look at your algorithm box in the appendix, I don't think he would notice the difference between the sampling distribution and discounted stationary distribution.
>
> I know it is super easier to modify the terminology to the correct one and say that, okay, we use the approximated sampling distribution as the discounted distribution. If so, why not use the undiscounted stationary distribution in your papers? For the Q-function and Bellman equation, it is often difficult to learn and define the undiscounted version, but techniques are almost the same for both discounted and undiscounted distributions to derive your algorithms. The undiscounted version might be simpler since there is no initial distribution now. And the intuition you mentioned in
>
> > In the following, we present some underlying reasons why RGM (sampling distribution sampling) could perform better in practice. In sampling distribution, every state-action pair along the episode is treated equally, while in discounted visitation distribution, we care more about the early state-action pairs than the state-action sampled far into the episode.
>
> would actually perfectly fit the undiscounted case. Why not do the undiscounted case?  I kindly agree with the argument with the discounted case. But things would actually hold in the undiscounted case.
>
> I am not asking you to perform more experiments, but I do believe a serious discussion and remark should be presented in your final or next version since the difference actually matters, especially when we really want to understand why discounted sometimes does not work and why we not just doing undiscounted case?
>
> -----
> 2.
>
> > Reward correction or reward shaping is far more difficult in offline setting than that in online setting.
>
> How do you draw the conclusion? Do you have any evidence support for the claim?
>
> To me, offline rl itself is indeed hard, but the offline RL benchmark (d4rl) is not hard.. Especially if you modify some reward functions to make it "not perfect".
>
> ----
>
> 3.
>
> >  Given a specific task, there may be one perfect reward that can describe this task.
>
> I kindly disagree. What if I multiply the reward function by a constant factor? all of them are perfect reward functions. It is really hard and ambiguous to define what is perfect for the mujoco dense reward function. To me, the perfect setting is still the sparse reward setting, the behavior of the policy would indicate what is perfect (completing the task).
>
> ----
>
> 4.
> For the typo, I mean the operator you applied on value function $V$, which only takes $s$ as input, which makes the notion a little bit ambiguous.
>
> ------------------
> I changed my score to 5, but personally, I still think the paper needs a major revision to address the concerns for both theoretical and suitable experimental settings.

---

> > ### Author Response · Authors · 2022-11-25
> > **Response to Q1**
> >
> > We would like to thank the reviewer for the thoughtful comments and for increasing the score! We really appreciate it! Here are responses to each comment:
> >
> > >**Q1. My opinion is based on your original paper draft. In the submitted version, you haven't distinguished the difference between discounted stationary (visitation) distribution...**
> >
> > - It should be noted that we did not modify or replace the terminology in our revision. Our definition of $d^{\pi_r}$ in Eq.(3) (Preliminaries Section) has always been the discounted state-action visitation distribution with $\gamma$, and is used throughout the paper. In our original paper, we use "state-action visitation distribution" for brevity, and to address the reviewer's concern in rebuttal phase 1, we add "discounted" before "state-action visitation distribution" to make it clearer in our updated version. It is common in RL community to consider the discounted setting, which enables better theoretical properties for convergence analysis.
> > - To further address the reviewer's concern, we additionally provide the theoretical derivation and the final form of the undiscounted version ($\gamma = 1$) of RGM, and explain how it will lead to a different form as compared to the original RGM:
> >     - When considering $\gamma = 1$, the RL objective (Eq.(1)) changes to (see [27] for discussion):
> >     $$
> >     {\pi}\_{r}^* = \underset{{\pi_r}}{\arg \max } \lim_{N \to \infty } \frac{1}{N+1} \mathbb{E} \left[\sum_{t=0}^{N} {r}\left(s_t, a_t\right) | s_0 \sim \mu_0(\cdot), a_t \sim {\pi_r}\left(\cdot | s_t\right), s_{t+1} \sim T\left(\cdot|s_t, a_t\right)\right]
> >     $$
> >     - Under certain conditions (in the discrete case, the MDP should be ergodic; in the continuous case, the conditions are more involved [28]), the undiscounted distribution $d^{\pi_r}$ is defined as the normalized distribution satisfying: $d^{\pi\_r}(s,a)=\mathcal{P}^{\pi\_r}\_{\star}d^{\pi\_r}(s,a),\forall s \in \mathcal{S},a \in \mathcal{A}$, where $\mathcal{P}^{\pi_r}\_{\star}$ is the transpose (or adjoint) policy transition operator, defined as $\mathcal{P}^{\pi_r}\_{\star}d^{\pi_r}(s,a)=\pi(a|s)\mathcal{T}\_{\star}d^{\pi_r}(s)$, please refer to [22] for more details.
> >     - Note that under the undiscounted setting $\gamma = 1$, the Bellman flow constraint Eq.(8) in RGM becomes ill-posed, since for any $d$ that satisfies the constraint Eq.(8) and any constant $c>0$, $d\leftarrow c\cdot d$ also satisfies the constraint Eq.(8). To address this issue, a typical treatment is to introduce an additional normalization constraint $\sum_{s,a}d(s,a)=1$ [13, 22, 27]. Thus the lower level problem Eq.(9) changes to:
> >     $$
> >     \begin{aligned}
> > d^{\pi^*\_{\hat{r}}} &= \underset{d \geq 0}{\arg \max}  \mathbb{E}\_{(s, a) \sim d}[\hat{r}(s,a)] - \alpha \text{D}\_f\left(d \| d^{\mathcal{D}}\right) \\\\
> > \text { s.t.} & \quad \sum_a d(s, a) = \mathcal{T}\_{\star} d(s), \sum_{s,a}d(s,a)=1, \forall s \in S
> > \end{aligned}
> >     $$
> >     - We introduce $\lambda$ as the Lagrangian multiplier of the additional normalization constraint $\sum_{s,a}d(s,a)=1$. By using analogous derivation as in Proposition 2, we could derive the undiscounted version of Eq.(13) as:
> > $$
> > \frac{d^{\pi^*_{\hat{r}}}(s,a)}{d^\mathcal{D}(s,a)}=f_{\star}^{\prime}\left(\frac{\hat{r}(s,a)+ \mathcal{T} V^*(s, a)-V^*(s)-\lambda^*}{\alpha}\right)
> > $$
> >     - Next, also by following similar derivations as in Eq.(23)-(41), the undiscounted bilevel optimization problem can be formulated as:
> >     $$
> >     \begin{aligned}
> >     \Delta r^* & = \arg \min\_{\Delta r} \mathbb{E}\_{(s,a) \sim d^{\mathcal{D}}}\left[w(s,a)f\left(f\_{\star}^{\prime}\left(\frac{\hat{r}(s,a)+ \mathcal{T} V^*(s, a)-V^*(s)-\lambda^*}{\alpha}\right)/w(s,a)\right)\right] \\\\
> >     s.t. \quad & V^*(s), \lambda^*=  \arg\min\_{V(s), \lambda} \alpha \mathbb{E}\_{(s,a)\sim d^{\mathcal{D}}}\left[f_{\star}\left(\frac{\hat{r}(s,a) + \mathcal{T}V(s,a)-V(s)-\lambda}{\alpha}\right)\right]+\lambda
> >     \end{aligned}
> >     $$
> >     - The downstream policy extraction and practical implementation are similar to that of discounted setting.
> > - The above is the anatomy of the undiscounted version for RGM. The undiscounted setting often poses some challenges for many RL algorithms because of the difficulty of defining Q-value and the convergence analysis of Bellman backup operators. Moreover, in the undiscounted version of RGM, we need to additionally optimize $\lambda$, which poses extra challenges and potential instabilities during the optimization process. Due to these drawbacks, we only presented the discounted version of RGM in original our paper.
> > - Thanks a lot for the review's comment, which greatly improves the clarity of our paper. We'll add the detailed derivations and discussions about the undiscounted version for RGM in our final version.

---

> > > ### Author Response · Authors · 2022-11-25
> > > **Response to Q2-Q4**
> > >
> > > > **Q2. How do you draw the conclusion? Do you have any evidence support for the claim? To me, offline rl itself is indeed hard, but the offline RL benchmark (d4rl) is not hard.. Especially if you modify some reward functions to make it "not perfect".**
> > >
> > > - Existing reward design/shaping approaches are only designed for online settings, which need to deploy the policy in the environment and use a trial-and-error process to adjust the reward and verify if the policy produces the desired behaviors. Such a paradigm is simply not possible under offline settings, as there is no way to directly verify the policies' performance without online interactions. Hence offline reward correction is far more challenging than the online case, as we need to simultaneously solve an offline RL problem (generally harder than online RL), as well as perform counterfactual reasoning on the resulting policies' performance without environment interactions. The latter does not present in online settings, which can be particularly hard.
> > > - Although the offline RL benchmark D4RL tasks may not be so hard, note that we are solving offline policy optimization problems with imperfect rewards, which are far more challenging than benchmark tasks. The severe performance drop of offline RL methods in Table 1 (main text) and Table 8 (Appendix E.3) all demonstrate the difficulty of such tasks.
> > >
> > > >**Q3. I kindly disagree. What if I multiply the reward function by a constant factor? all of them are perfect reward functions. It is really hard and ambiguous to define what is perfect for the mujoco dense reward function. To me, the perfect setting is still the sparse reward setting, the behavior of the policy would indicate what is perfect (completing the task).**
> > >
> > > - As discussed in a number of prior studies [23-25], an optimal or perfect reward should **define both the task the agent learns to solve, as well as the "bread crumbs" that allow agents to efficiently learn to solve the task**. Please refer to Section 2.1 of [23] for a thorough discussion on the properties of perfect rewards. There should be no reward gap if we train and test using the same reward function, as we are trying to learn and evaluate in the same task. However, in our paper, we are trying to solve the corresponding tasks of the original D4RL rewards given only some corrupted (imperfect) rewards.
> > >
> > > - **Multiplying the reward function by a constant factor in fact also impacts policy learning when using function approximators**, as discussed in the "Perfect Rewards" part of Related Work in our submitted paper. Briefly, a recent paper [26] studies this phenomenon and finds that the linear transformation of rewards is equivalent to engineering the initialization of Q-function estimation. A positive reward shifting leads to conservative exploitation, while a negative reward shifting leads to curiosity-driven exploration [26]. Hence an improperly scaled reward function may act similarly to an imperfect reward that leads to over-conservative or over-aggressive sub-optimal policies. Only a properly scaled reward function can guarantee the best policy learning performance.
> > > - The sparse reward setting mentioned by the reviewer is actually a type of imperfect reward, as reward signal is only provided until reaching the goal condition, while no learning signal is provided before reaching the goal to facilitate efficient policy learning. We have actually validated our method on several sparse reward tasks. In our latest revision, We have included additional experiments on the difficult Antmaze sparse reward tasks. Table 13 shows that RGM is the only offline RL method that can solve such extremely difficult sparse reward tasks. In addition, the 8X8 grid world navigation task in Section 5.3 and Appendix C.2 is also a sparse reward task.
> > >
> > > >**Q4. For the typo, I mean the operator you applied on value function $V$, which only takes $s$ as input, which makes the notion a little bit ambiguous.**
> > > - We thank the reviewer for this detailed comment. However, it is worth mentioning that the transition operator $\mathcal{T}V$ should take $(s,a)$ as input instead of $s$ only. Because $\mathcal{T}V(s,a)$ by our definition in the paper means the expectation of $V$ at the next state $s'$ that **transferred from $(s,a)$**,  i.e.,$\mathcal{T}V(s,a)=\sum_{s'}T(s'|s,a)V(s')$.
> > > - Because the phase 1 discussion has already finished and we are no longer allowed to revise the paper, we'll keep it in mind and clarify it more clearly in our next version. Thank you!

---

> > > > ### Author Response · Authors · 2022-11-25
> > > > **Additional References**
> > > >
> > > > [23] David Abel, Will Dabney, Anna Harutyunyan, Mark K Ho, Michael Littman, Doina Precup, and Satinder Singh. On the expressivity of markov reward. Advances in Neural Information Processing Systems, 34:7799–7812, 2021.
> > > >
> > > > [24] Satinder Singh, Richard L Lewis, and Andrew G Barto. Where do rewards come from? In Proceedings of the Annual Conference of the Cognitive Science Society, 2009.
> > > >
> > > > [25] Jonathan Sorg. The Optimal Reward Problem: Designing Effective Reward for Bounded Agents. PhD thesis, University of Michigan, 2011.
> > > >
> > > > [26] Hao Sun, Lei Han, Rui Yang, Xiaoteng Ma, Jian Guo, and Bolei Zhou. Optimistic Curiosity Exploration and Conservative Exploitation with Linear Reward Shaping . In Advances in Neural Information Processing Systems, 2022.
> > > >
> > > > [27] Zhang, Ruiyi, et al. "GenDICE: Generalized Offline Estimation of Stationary Values." International Conference on Learning Representations. 2019.
> > > >
> > > > [28] Meyn, Sean P., and Richard L. Tweedie. Markov chains and stochastic stability. Springer Science & Business Media, 2012.

---

### Author Response · Authors · 2022-11-12
**Revision Summary**

We thank all the reviewers for the detailed and constructive comments. We have revised the paper to address the concerns of the reviewers. The summary of changes in the updated version of the paper is as follows:

1. We add the experiments on sampling from discounted distribution in Appendix E.2.
2. We add the comparisons with SOTA offline RL methods under completely incorrect rewards in Appendix E.3.
3. We add experiments on different degrees of noisy imperfect rewards in Appendix E.4.
4. We add ablations on the number of expert trajectories for RGM, DWBC and SMODICE in Appendix E.5.
5. We add experiments under sparse reward setting in Appendix E.6.
6. We add an illustration for the learned rewards under continuous MDPs in Appendix E.7.
7. We add a run time comparison of RGM and other baselines in Appendix B.3.
8. We add an illustrative example of the reward differences between RGM and SMODICE/DemoDICE in Appendix C.2.
9. We add RGM(T) in Table 1 in the main text.
10. We revise the description of Figure 5 to make it more clear and give the explanations of "-r/-m/-m-r/-m-e".
11. We add a new section in Appendix F to discuss the adaptation of RGM to the online setting.

---

### Author Response · Authors · 2022-11-16
**A Kind Reminder**

Dear reviewers,

It's a kind reminder that there are 72 hours left for the discussion 1 phase. Please let us know if our corrections, clarifications, and additional results have addressed your concerns. If you have further concerns or questions, we are very glad to address them.

---

### Author Response · Authors · 2022-12-04
**A Kind Reminder in Phase 2**

Dear reviewers and AC,

We're terribly sorry to bother you with this reminder. However, it's a kind reminder that about only one week is left for the rebuttal. We think that our responses have already addressed most of the reviewers' concerns and **we will be more than happy to address your further concerns**. We understand you may be very busy due to your work and the overlaps with NeurIPS, so we really appreciate your time engaged in the review and rebuttal phase.

Thank you very much and look forward to your replies!

Authors of Paper 448

---

### Decision · Program_Chairs · 2023-01-20

**Decision:**

Accept: poster

**Justification For Why Not Higher Score:**

The scope of the problem is not broad enough.

**Justification For Why Not Lower Score:**

This paper studies a novel problem, and the proposed approach is interesting.

**Metareview: Summary, Strengths And Weaknesses:**

This paper studies the offline RL with imperfect rewards and proposes an approach, RGM, to tackle this problem. The proposed approach is mathematically intuitive and empirically effective. The reviewers raised concerns about writing and technical derivations. The AC thanks the reviewers and authors for their extensive discussions.
Most of the concerns are addressed during the rebuttal period. The AC thus recommends acceptance. The AC also highly encourages the authors to polish the paper further and address all remaining concerns in the final version.

**Note From Pc:**

if the above contains the word "oral" or "spotlight" please see: "oral" presentation means -> notable-top-5% and "spotlight" means -> notable-top-25%. As stated in our emails, we are disassociating presentation type from AC recommendations